# Observed Trends of Clouds and Precipitation (1983–2009): Implications for Their Cause(s)

Xiang Zhong[1], Shaw Chen Liu[1], Run Liu[1], Xinlu Wang[1,2], Jiajia Mo[1], and Yanzi Li[1]

[1]Institute for Environmental and Climate Research, Jinan University, Guangzhou, 511486, China
[2]Hangzhou AiMa Technologies, Hangzhou, 311121, China

*Correspondence to*: Shaw Chen Liu (shawliu@jnu.edu.cn) and Run Liu (liurun@jnu.edu.cn)

**Abstract.** Satellite observations (International Satellite Cloud Climatology Project (ISCCP), 1983–2009) of linear trends in cloud cover are compared to those in global precipitation (Global Precipitation Climatology Project (GPCP) pentad V2.2, 1983–2009), to investigate possible cause(s) of the linear trends in both cloud cover and precipitation. The spatial distributions of the linear trends of total cloud cover and precipitation are both characterized primarily by a broadening of the major ascending zone of Hadley circulation. Our correlation studies suggest that global warming, Atlantic Multidecadal Oscillation (AMO), and Pacific Decadal Oscillation (PDO) can explain 67%, 49% and 38%, respectively, of the spatial variabilities in the linear trends in cloud cover but causality is harder to establish. Further analysis of the broadening of the major ascending zone of Hadley circulation shows that the trend in global temperature, rather than those of AMO and PDO, is the primary contributor to the observed linear trends in total cloud cover and precipitation in 1983–2009. The underlying mechanism driving this broadening is proposed to be the moisture–convection–latent heat feedback cycle under global warming conditions. The global analysis is extended by investigating connections between clouds and precipitation in China, based on a large number of long-running, high-quality surface weather stations in 1957–2005. This reveals a quantitative matching relationship between the reduction in light precipitation and the reduction in total cloud cover. Furthermore, our study suggests that the reduction in cloud cover in China is primarily driven by global temperature; PDO plays a secondary role, while the contribution from AMO and Niño3.4 is insignificant, consistent with the global analysis.

## 1 Introduction

Long term changes in cloud cover are of great importance to the climate as well as the entire ecosystem. Changes in cloud cover associated with climate change remain one of the most challenging aspects of predicting future climate change. Previous studies have shown that over land, except for the Arctic, central northern Africa and the Pacific islands around Indonesia, show various decreasing trends (Schulz et al., 2011; Eastman and Warren, 2013; Free and Sun, 2013; Norris et al., 2016; Rajeevan and Nayak, 2017Mahlobo et al., 2019). In China there are a number of studies reporting a significant decrease in total cloud cover ranging from -0.76% per decade to -0.9% per decade during the past few decades (Kaiser, 1998, 2000; Liang and Xia, 2005; Xia, 2010; Xia, 2012; Liu Y. et al., 2016). Over the ocean, the equatorial central Pacific and midlatitudes of both

hemispheres, northern Atlantic, and places around Australia show also a decreasing trend. On the other hand, the tropical western Pacific, the subtropical eastern Pacific of both hemispheres, southern Atlantic, and nearly the entire Indian Ocean show increasing trends (Chen et al., 2019; Mao et al., 2019).

In a study of changes in cloud cover observed from land stations worldwide (1971–2009), Eastman and Warren (2013) found that global average trends of cloud cover suggest a small decline in total cloud cover, on the order of 0.4% per decade. Their analysis of zonal cloud cover changes suggests widening tropical belt and poleward shifts of the jet streams in both hemispheres associated with global warming. In addition, they found that changes in cloud types associated with the Indian monsoon are consistent with the suggestion of black carbon aerosols affecting monsoonal precipitation, causing drought in northern India. On the other hand, they found that northern China, where large emissions of anthropogenic aerosols exist, did not show an obvious aerosol connection. Norris et al. (2016) showed that several independent, empirically corrected satellite records exhibit large-scale patterns of cloud change between the 1980s and the 2000s that are similar to those produced by model simulations of climate with recent historical external radiative forcing. Observed and simulated cloud change patterns are consistent with poleward retreat of mid-latitude storm tracks, expansion of subtropical dry zones, and increasing height of the highest cloud tops at all latitudes. The primary drivers of these cloud changes appear to be increasing greenhouse gas concentrations and a recovery from volcanic radiative cooling. These findings are consistent in general with those of Eastman and Warren (2013).

Chen et al. (2019) investigated changes in clouds associated with decadal climate oscillations including the Pacific decadal oscillation (PDO) and the Atlantic multidecadal oscillation (AMO) by comparing cloud cover data (1983–2009) over the oceans from the International Satellite Cloud Climatology Project (ISCCP) (Schiffer and Rossow, 1983) with General Circulation Models (GCM) simulations. They found that the observed linear trends in cloud cover are more closely related to decadal variability (including PDO and AMO) than to greenhouse gases (GHG) induced warming. It should be noted that the changes/trends in cloud cover over the oceans found in Chen et al. (2019) are in good agreement with those of Eastman et al. (2011), which were derived from synoptic observations made by observers on ships. The agreement provides credence to both data sets and the major patterns of the changes/trends in cloud cover derived in the two studies. On the other hand, the two studies differ on attributing the trends of cloud cover to global warming, PDO and/or AMO. In this context, we note that PDO, AMO and global temperature all have significant linear trends during the relatively short period 1983–2009 studied by Chen et al. (2019), while PDO did not have any trend during the period 1971–2009 studied by Eastman and Warren (2013).

Closely related to the changes in cloud cover, there are extensive reports of enhancements in heavy precipitation and reductions in the light and moderate precipitation in China (Karl and Knight, 1998; Manton et al., 2001; Klein Tank and Können, 2003; Sun et al., 2007; Wang and Zhai, 2008; Liu et al., 2009; Shiu et al., 2012; Wu and Fu, 2013; Jiang et al., 2014; Liu et al., 2015; Liu R. et al., 2016), as well as in a widespread land and oceanic areas around the globe (Karl and Knight, 1998; Manton et al., 2001; Klein Tank and Können, 2003; Fujibe et al., 2005; Groisman et al., 2005; Goswami et al., 2006; Adler et al., 2017). These changes in precipitation extremes have been attributed primarily to global warming (Trenberth, 1998; Allen and Ingram, 2002; Liu et al., 2015). Trenberth et al. (2003) summarized the global warming theory as follows. In the

global warming environment, if everything else remains the same, the precipitation intensity of a storm should increase at the same rate as the atmospheric moisture which increases at about 7% $K^{-1}$ according to Clausius–Clapeyron (C–C) equation. They further argued that the increase in heavy rainfall can even exceed 7% $K^{-1}$ because additional latent heat released from the increased water vapor can invigorate the storm and pull in more moisture from the boundary layer. This forms a positive moisture–convection–latent heat feedback cycle (hereafter referred to as MCL–Feedback cycle). An invigorated storm (i.e. heavy precipitation) can remove more moisture than the C–C value from the atmosphere, leaving less than the C–C moisture available for light and moderate precipitation (Trenberth et al., 2003). In this context, Liu R. et al. (2016) found that as the climate warms there are extensive enhancements and expansions of the three major tropical precipitation centers–the Maritime Continent, Central America, and tropical Africa–leading to the observed widening of Hadley cells and a significant strengthening of the global hydrological cycle (Reichler and Held, 2005; Hu and Fu, 2007; Zhou et al., 2011; Davis and Rosenlof, 2012; Eastman and Warren, 2013; Norris et al., 2016).

There is a strong relationship between precipitation extremes and cloud top temperature (Arkin and Meisner, 1987; Kuligowski, 2002; Lau and Wu, 2011). Lau and Wu (2011) investigated the climatological characteristics of tropical rain and cloud systems over Tropics using the brightness temperature (BT) data obtained from Visible and Infrared Scanner (VIRS) and the precipitation data gathered from Tropical Rainfall Measuring Mission (TRMM) Microwave Imager (TMI) and Precipitation Radar (PR). It is found that the top 10% heavy precipitation appears to be associated with high cloud tops and light precipitation has a close association with low clouds.

In this study, we first examine the worldwide satellite observations (ISCCP, 1983–2009) of changes in cloud cover. These changes are compared to changes in global precipitation (Global Precipitation Climatology Project, GPCP pentad V2.2, 1983–2009), and the results are used to decipher possible cause(s) of the changes in both cloud cover and precipitation. To our knowledge, no previous paper has analysed changes in both clouds and precipitation. We then examine the reduction in cloud cover in China. Taking advantage of the extensive daily observations of cloud cover and precipitation from Chinese surface meteorological stations over a relatively long period (1957–2005), we will try to establish a quantitative matching relationship between changes in cloud cover and precipitation. The rest of this paper is organized as follows: data and methodology are presented in Sec. 2, results in Sec. 3, and a summary and conclusions in the final section.

## 2 Data and methodology

Cloud cover from ISCCP during 1983–2009 (2.5°×2.5°, monthly) is used in this study. To get rid of the influence of artifacts from changing satellite view angles, changing solar zenith angles, and other sources of spurious trends in the records, an empirical method is applied (Norris and Evan, 2015). By removing anomalous cloud variability within individual grid boxes shown to be associated with artifact factor anomalies, the spatial anomalies relative to an unknown global mean value are left. We use the annual anomalies of total cloud cover to get the spatial distribution of long term trends (https://rda.ucar.edu/datasets/ds741.5/). Precipitation data from GPCP (V2.2, 1983–2009, 2.5°×2.5°, pentad) are used in this

study (Xie et al., 2003). The dataset is available from National Oceanic and Atmospheric Administration (NOAA) National Climatic Data Center (NCDC) at ftp.ncdc.noaa.gov/pub/data/gpcp.

In addition, annual average of global temperature anomaly from NCDC (https://www.ncdc.noaa.gov/cag/global/time-series/globe/land_ocean/ann/12/1957-2005) and PDO, AMO, Niño3.4 from NOAA Working Group on Surface Pressure
(WGSP, http://psl.noaa.gov/gcos_wgsp/Timeseries/) are also used. The PDO is defined by the leading EOF mode of the monthly anomalous sea surface temperature (SST) in the North Pacific (poleward of 20° N) with global mean SST anomaly subtracted. To make sure AMO also gets rid of the influence from global mean SST, we did a revision using method provided by Trenberth and Shea (2006). The high frequency signals of both indexes are retained in this study.

Daily cloud cover and precipitation data of Chinese surface meteorological stations from China Meteorological Data Service
Center are also analyzed to examine the reduction in cloud cover in China (http://data.cma.cn/site/index.html). To ensure the consistency and integrity of the data and because data on the cloud cover is available only up to 2005, we select 477 surface meteorological stations and set 1957–2005 as the study period. Total number of samples in each station are required to have less than 5% missing data during the studied periods, and each station has at least 17002 $[(37 \times 365 + 12 \times 366) \times 95\%]$ valid records of both precipitation and cloud cover. The number of valid records in each station has a temporal variation, but results
for stations selected by a much stricter standard (annual missing days ≤5 days) highly support which of the 477 stations (not shown). Figure S1 shows the spatial distribution of the stations.

Linear regression between two atmospheric parameters is evaluated by a traditional scatter correlation method. Ten categories of precipitation with increasing intensities are calculated by dividing the 49 years (1957–2005) average spectrum of precipitation into ten categories with equal precipitation amount. Some words of caution are due here that precipitation data
from all 477 stations use the same thresholds for sorting different intensity categories in this study. The ranges of the 10 bins for the period of 1957–2005 are 4.0, 7.6, 11.6, 16.1, 21.4, 28.1, 37.1, 50.7, 76.4 and ≥76.4 mm day$^{-1}$. The test of significance used in this study is student's $t$ test. When testing the correlation between two spatial distributions, we followed the method introduced in Bretherton et al. (1999) to calculate the effective sample size.

## 3 Results

### 3.1 Regional trends of cloud cover and precipitation

Figure 1a shows the linear trends in total cloud cover (ΔTCC/Δt) derived from corrected ISCCP D2 data set (1983–2009). The general pattern of trends over the oceans are in excellent agreement with those reported by Chen et al. (2019). The pattern is also consistent with those derived by Eastman and Warren (2013) from Extended Edited Cloud Reports Archive (EECRA) data set for 1971–2009. The linear trends of annual total precipitation (ΔTP/Δt) derived from GPCP pentad V2.2, (1983–2009)
are shown in Fig. 1b. These trends are in good agreement with results of previous studies (Zhou et al., 2011; Liu R. et al., 2016). The climatological average annual precipitation rates are shown in green contours in both Figs. 1a and 1b to facilitate comparison between the patterns of clouds and precipitation.

There is a high degree of consistency between the general patterns of Figs. 1a and 1b. This can be seen by first noticing a prominent feature of a loose circle of warm color patches (increases in cloud cover and precipitation) in both Figs. 1a and 1b centered around Kalimantan, Indonesia, starting in northern Australia circling counter clockwise to the Philippines, to western China, turning southward along western Indian Ocean all the way to about 50° S, covering nearly half of the eastern hemisphere (0° E–180° E). This loose circle of warm color patches exists, albeit not at the exactly same location, in both Figs. 1a and 1b, and has an obvious effect of broadening the center of precipitation (ascending/wet zone of Hadley cells) over the Maritime Continent in all directions (Zhou et al., 2011; Liu R. et al., 2016). There are also significant and extensive enhancements/broadenings of precipitation centers over Central America and equatorial Africa. These enhancements/broadenings have been interpreted to be an essential part of the widening of Hadley circulation and poleward shifts of the jet streams associated with global warming (Reichler and Held, 2005; Hu and Fu, 2007; Zhou et al., 2011; Davis and Rosenlof, 2012; Eastman and Warren, 2013; Liu R. et al., 2016; Norris et al., 2016).

As a measure of the broadening of the major ascending zone of Hadley circulation, we calculate and illustrate the expansion of cloud cover and precipitation as a function of 16 rectangular belts centered in the middle of Kalimantan, Indonesia which is located near the major ascending/wet zone of Hadley cell (Fig. 2). Each rectangular belt is 2.5 degree wide in both latitude and longitude except the first rectangle is 5 degree wide in latitude and 55 degree wide in longitude. The summing up was done for each rectangle independently, the inner rectangles were not included. Figures 3a and 3e depict for annual precipitation and total cloud cover, respectively, their "climatology" (black curve) and "climatology + change during 1983–2009" (blue curve). It can be seen that, for a specific value of the y-axis, the blue curve is characterized by a shift horizontally (x-axis direction) to the right (i.e. higher number of belt) of the black curve for most of Figs. 3a and 3e. In comparison, there is very little upward shift in the vertical or y-axis direction, especially at low end (belt 1 and 2) and high end belts (belt 15 and 16). As a result, there is hardly any enhancement in total cloud cover and total precipitation. These characteristics can be interpreted as an expansion to higher latitudes and wider longitudes, i.e. broadening of the major ascending zone of Hadley circulation, but without any significant enhancement in total cloud cover or total precipitation during the period of 1983–2009. Remarkably one can see that the expansion of the major ascending zone of Hadley cell as measured by clouds (Fig. 3e) starts at belt 2 (3.75° latitude). Meanwhile, the expansion of the major ascending zone of Hadley cell as measured by precipitation (Fig. 3a) starts at belt 5 (12.5° latitude). The reason for the difference is unknown, one possible reason could be the constraint on the total annual precipitation, which is equal to global evaporation and determined by the global surface energy budget, increases with global temperature at a rather small rate of about 2%–3% $K^{-1}$ (Cubasch et al., 2001). Quantitatively the degree of expansion depends on the selected value of the y-axis, increasing quickly when the value is near 1000mm precipitation level (Fig. 3a) or 55% of TCC (Fig. 3e). The value of the shift is typically within the range of one quarter to three quarters of a belt width (2.5 degree), or about 0.6–1.9 degree. These annual values are comparable to the poleward shift of the subtropical dry zones (up to 2° decade$^{-1}$ in June-July-August (JJA) in the Northern Hemisphere and 0.3–0.7° decade$^{-1}$ in June-July-August and September-October-November in the Southern Hemisphere) found by Zhou et al. (2011). As a summary of this and the last paragraphs, a logical conclusion can be drawn that the linear trends of TCC and TP are mainly characterized by a broadening of the major

ascending zone of Hadley circulation in both latitude and longitude associated with global warming. These characteristics will be used in the following as key criteria for the evaluation of relative contributions of individual climate indexes to the linear trends in total cloud cover (TCC) and total precipitation (TP).

A critical question remaining is whether climate parameters other than global warming may also contribute to the linear trends of TCC and TP, particularly AMO and PDO, which also have significant linear trends (Fig. S2) during the relatively short period of 1983–2009 of this study? This is addressed in the following. Figures 4a and 4b depict slopes of linear regressions of TCC and TP against the annual average global temperature anomaly (GT) at each grid point, i.e. $\Delta TCC/\Delta GT$ and $\Delta TP/\Delta GT$, respectively. The circle of warm-color patches around Kalimantan, which symbolizes the broadening of the major ascending zone of Hadley circulation, is highly visible in both Figs. 4a and 4b. The broadening as a function of the 16-rectangular belts is illustrated in Figure 3b for TP and Fig. 3f for TCC. The enhancement/broadening of the precipitation centered over equatorial Africa can also be clearly seen. These agreements between Figs. 1a and 4a; and between Figs. 1b and 4b are substantiated by the relatively high correlation coefficients of 0.82 and 0.93, respectively (Table 1), suggesting that global warming can explain 67% and 86%, respectively, of the spatial variabilities of linear trends in cloud cover and precipitation. It is important to note that these high degrees of correlation do not establish a causal relationship; they merely suggest there are probabilities of finding a causal relationship. The slopes of linear regressions of TCC and TP against AMO at each grid point are plotted in Fig. 5a and 5b. The general patterns also match reasonably well with those of Figs. 1a and 1b, respectively. The correlation coefficients between Figs. 1a and 4a, and between Figs. 1b and 4b, are slightly less than those of global temperature at 0.70 and 0.77, respectively (Table 1), implying that AMO can explain about 49% and 59%, respectively, of the spatial variabilities of linear trends in cloud cover and precipitation. Due to limited space, figures for PDO and Niño3.4 corresponding to Figs. 4a and 4b are presented in the Supplement (Figs. S3 and S4). The correlation coefficients of Niño3.4 with Figs. 1a and 1b are very low at -0.20 and 0.02, respectively (Table 1). A major reason for the low correlation coefficient is the lack of any linear trend in Niño3.4 during 1983–2009. Correlation coefficients of PDO with Figs. 1a and 1b are 0.62 and 0.73 (Table 1), respectively, slightly less than those of AMO. These correlation coefficients imply that PDO can explain approximately 38% and 53%, respectively, of the spatial variability of linear trends in cloud cover and precipitation.

It should be noted that considering PDO together with AMO and GT, there obviously is a problem of over 100% explanation of the spatial variabilities of linear trends in cloud cover and precipitation. Since the trend of global SST has been removed from the PDO and AMO indexes in this study, in theory the trend in GT should be orthogonal to those of PDO and AMO. In practice the orthogonality is not attained because the trend of global SST doesn't equal the real influence of GT on PDO or AMO. It is difficult to completely remove the influence of GT from PDO or AMO index, which is likely the main reason of the problem of over 100% explanation. Table 1 presents the correlation coefficients with Figs. 1a and 1b for various linear combinations of GT and other three climate indexes. Significant improvements of the correlation with Fig. 1a (TCC) are attained when GT is paired with AMO (0.86) or Niño3.4 (0.89). The correlation with Fig. 1b (TP) is not improved by any combination, which is understandable as the correlation coefficient of GT alone (0.93) is already very high.

Table 1 has also been evaluated for detrended data of TCC, TP, GT, AMO, PDO and Niño3.4 (Table S1). The correlation coefficients are all less than 0.33, implying that consecutive yearly variabilities contribute insignificantly to the high correlation coefficients in Table 1, and the high correlation coefficients are nearly entirely contributed by the long-term linear trends of GT on PDO and AMO. One of the reasons for the lack of correlation could be due to the small consecutive yearly variabilities relative to the long-term linear trends (about 1 to 10) for GT, PDO and AMO (Fig. S2).

Based on Table 1, we can conclude that the linear trends of GT, AMO and PDO all have a good probability in contributing to the observed linear trends of total cloud cover and precipitation in 1983–2009. However, the results of Table 1 do not provide any clue indicating which one of the three is the primary contributor. To address this question, we examine Figs. 3b-3d and 3f-3h to evaluate how do the trends of GT, AMO and PDO influence the observed linear trend of broadening of the major ascending zone of Hadley circulation as shown in precipitation and total cloud cover in 1983–2009. Figs. 3b-3d show

the changes (blue curve) from the climatology (1983–2009) (black curve) in the annual total precipitation (mm) of the 16 belts of Fig. 2 as a function of GT, AMO and PDO, respectively. The formula for calculating the blue curve, for instance for the changes in precipitation as a function of global temperature (Fig. 3b), is $d(TP)/d(GT) \times \Delta GT$, where $\Delta GT$ denotes difference in the global temperature between 1983 and 2009. It can be seen that Fig. 3b (GT) agrees very well with Fig. 3a both qualitatively and quantitatively; while Figs. 3c and 3d have significantly greater positive values (broadening) compared to the small negative

values (contraction) of Fig. 3a for the inner 5 belts, resulting in a significant enhancement of the overall precipitation. This discrepancy is crucial, as the quantity of global total annual precipitation, which is equal to global evaporation and determined by the global surface energy budget, increases with global temperature at a rather small rate of about 2%–3% $K^{-1}$ (Cubasch et al., 2001), which is manifested in Figs. 3a and 3b by the small/negligible change of the net area between blue and black lines, while both Figs 3c and 3d have significant enhancements. Therefore, based on the results of Figs. 3a-3d, we propose that the

trend in global temperature, rather than that of AMO and PDO, is the primary contributor to the observed linear trend of precipitation in 1983–2009. Likewise, Figs. 3e and 3f both have small/negligible changes of the net areas between blue and black lines, while both Figs 3g and 3h have significant enhancements of cloud cover. Accordingly, we propose that the trend in global temperature, rather than that of AMO and PDO, is the primary contributor to the observed linear trend of cloud cover in 1983–2009.

A comparison of Fig.1b with Fig. 3e reveals that the enhancements in precipitation in the tropics (Fig. 1b) are the major contributor to the broadening of the major ascending zone of Hadley circulation in observed precipitation (Fig. 3e). Since it has been shown by Liu R. et al. (2016) that the enhancements in precipitation in the tropics are nearly entirely driven by heavy precipitation (strong convections), we propose that the broadening of the major ascending zone of Hadley circulation is primarily driven by the moisture-convection-latent heat feedback cycle.

In summary of Sec. 3.1, the spatial distributions of the linear trends of total cloud cover and precipitation are characterized primarily by a broadening of the center of precipitation (ascending/wet zone of Hadley cells) over the Maritime Continent in all directions (Zhou et al., 2011; Liu R. et al., 2016). Our correlation studies show that GT, AMO and PDO can each explain significant spatial variabilities of the linear trends in cloud cover (67%, 49% and 38%, respectively) and precipitation (86%,

59% and 53%, respectively). Contribution by Niño3.4 itself is insignificant because it doesn't have any trend in 1983–2009. A linear combination of GT and AMO can explain as much as 74% and 79%, respectively, of the spatial variabilities of linear trends in cloud cover and precipitation. Further analysis of the broadening of the major ascending zone of Hadley circulation (Figs. 3a-3h) shows that the trend in global temperature, rather than that of AMO and PDO, is the primary contributor to the observed linear trend of total cloud cover and precipitation in 1983–2009. The underlying mechanism driving this broadening is proposed to be the moisture–convection–latent heat feedback cycle under global warming conditions.

The effects of anthropogenic aerosols on clouds and precipitation by acting as cloud condensation nuclei (CCN) is a highly controversial issue, which has been discussed extensively in a number of studies as well as one of our earlier papers (Liu et al., 2015). We defer the discussion on this issue to future studies, and acknowledge here that the CCN effects could introduce an unknown degree of uncertainty in this study. The long-term radiative effect of aerosols on the global temperature and other climate parameters are expected to be imbedded in the observed changes of these climate parameters, and thus included in this study.

Our results suggest that the global temperature contributes the most to the trends of cloud cover is more in line with the view of Eastman and Warren (2013), rather than with that of Chen et al. (2019) who suggested that AMO and PDO contributed more than the global temperature. However, it is well known that correlation method does not imply any cause–effect relationship, certainly not quantitative cause–effect relationship. Our analysis in this section have used correlation method, so were the study by Chen et al. (2019) and many studies on attributing the broadening of the major ascending zone of Hadley circulation to global warming cited by Eastman and Warren (2013). In this context, we note that Eastman and Warren's analysis covered a longer period 1971–2009 in which PDO did not have any significant linear trend, and hence could not have any contribution to the linear trends of cloud cover. This conclusion which does not rely on correlation method should override those derived from correlation studies, including those associated with PDO in this section and those derived by Chen et al. (2019).

### 3.2 Trends of cloud cover and precipitation from station data in China

The global analysis is extended in this section by investigating connections between clouds and precipitation in China, where is a large number of long-running, high-quality surface weather stations over the period of 1957–2005. The long-running data enable the analysis to be carried out over a period in which the linear trends of AMO and PDO have both diminished to insignificant values. More importantly, the high-quality data allow us to make meaningful analysis without using the correlation method, which has an intrinsic weakness because it confuses associations between variables and causal relationships as discussed above.

Data on cloud cover and precipitation from 477 surface meteorological stations provide significant higher spatial and temporal resolution and over longer time period (1957–2005 for TCC, 1957–2017 for TP) than satellite data, such that detailed analysis can be carried out to reveal fine features for different periods of time. Figure 6 shows linear trends in annual precipitation amount (ΔP) falling within each of the ten bins of equal rain rate with increasing precipitation intensity during

1957–2005. There is a significant overall shift toward higher precipitation intensity, in agreement with previous studies (Liu et al., 2005; Zhai et al., 2005; Sun et al., 2007; Qian et al., 2007; Wang and Zhai, 2008; Liu et al., 2009; Shiu et al., 2012; Wu and Fu, 2013; Jiang et al., 2014; Liu et al., 2015; Liu R. et al., 2016). Specifically, the bottom 10% light precipitation decreases by -1.5 ± 0.5 % per decade and the top 10% heavy precipitation increases by 2.7 ± 1.0 % per decade, both significant at the 99% confidence level. These values are robust over different time periods, for example for overlapping period (1983–2009) with satellite data, the bottom 10% light precipitation decreases by -2.8 ± 1.7 % per decade and the top 10% heavy precipitation increases by 8.0 ± 3.4 % per decade, the latter is significant at the 95% confidence level. For the period 1957–2017, the bottom 10% light precipitation decreases by -2.0 ± 0.4 % per decade and the top 10% heavy precipitation increases by 3.0 ± 0.7 % per decade, both significant at the 99% confidence level.

The linear trend in the non–precipitation days is 4.5 ± 0.2 days per decade, which is significant at the 99% confidence level (Table 2 and Fig. 7a). During the 49 year period, non–precipitation days have increased by about 22 days, which is nearly completely compensated by the decrease in light precipitation days. The bottom 10% precipitation alone has decreased by about 21 days, accounting for ~95% of the change of non–precipitation days. Note that the value of the bottom 10% precipitation days in Fig 7 has been multiplied by -1 to better compare with the non-precipitation days; the same is true for the bottom 10%~30% precipitation days. This value quickly approaches 100% when changes in the bottom 10%–40% precipitation days are included. This is fully expected as the number of bottom 40% precipitation days (147) account for ~90% of total precipitation days (163). In the meantime, the top 60% precipitation days barely change.

During the 49 year period, the number of cloud–free days has increased by about 11 days, accounting for one half of the increase in non–precipitation days (Fig. 7b and Table 2). This value quickly approaches 21 days when changes in the (0–50)% cloud cover days (CCD) are included. Twenty one days account for 95% in the increase of non–precipitation days. This is reasonable as precipitation usually does not occur when the cloud cover is less than 50%. Linear trends in the cloud–free days (CFD) and CCD are 2.3 ± 0.1 and 4.3 ± 0.2 days per decade, respectively, both significant at the 99% confidence level (Table 2). This is compensated by a reduction of 50%–100% in cloud cover days (Fig. 8), mostly by the 90–100% overcast days. This is also logical because precipitation tends to occur when the sky is heavily overcast (see also Fig.1). Since light precipitation days account for most of precipitation days, their decrease should approximately equal the decrease in overcast days.

So far in Sec. 3.2, we have used observed cloud cover and precipitation data from Chinese surface meteorological stations to successfully establish a quantitative matching relationship starting from the reduction in light precipitation days, to the increase of non–precipitation days, then to the increase in cloud free days and finally to the reduction of total cloud cover in China. This relationship is established via a straightforward arithmetic analysis, which is more robust than the correlation analysis, as the correlation analysis tends to introduce extra uncertainties (e.g. the causal relationship problem) as discussed in the last section. A critical remaining question is what is the cause of the reduction in light precipitation days in China? Liu et al. (2015) proposed that the reduction in light precipitation days in China is part of the extensive worldwide reports of enhancements in heavy precipitation and reductions in the light and moderate precipitation (Karl and Knight, 1998; Manton et al., 2001; Klein Tank and Können, 2003; Fujibe et al., 2005; Groisman et al., 2005; Liu et al., 2005; Zhai et al., 2005; Goswami

et al., 2006; Qian et al., 2007; Sun et al., 2007; Wang and Zhai, 2008; Liu et al., 2009; Wu and Fu, 2013; Jiang et al., 2014; Shiu et al., 2012; Liu et al., 2015; Liu R. et al., 2016); and the primary driving mechanism is the MCL–Feedback cycle under global warming environment proposed by Trenberth et al. (2003). We check this proposal by making the following evaluation of the trend in the bottom 10% light precipitation (B10LP) using its slope of linear regression against various climate oscillation indexes. For example, the trend of B10LP can be calculated from the trend of PDO as the following:

Calculated trend of B10LP from PDO for 1957–2005 = ($\Delta$B10LP/$\Delta$PDO) × (trend of PDO) = (−0.33 ± 0.09)% per decade. Where $\Delta$B10LP/$\Delta$PDO is the slope of linear regression between B10LP and PDO during 1957–2005. This calculated trend should be interpreted as the maximum possible contribution to the trend of B10LP from PDO, because there may be other climate parameters contributing to the slope ($\Delta$B10LP/$\Delta$PDO). Table 3 lists the trends of B10LP calculated from PDO, AMO and GT for three time periods of interest in this study: 1957–2005, 1957–2017 and 1983–2009. Niño3.4 is not listed because it has no linear trend during these periods and thus by definition no significant contribution.

Calculated trends of B10LP from GT agree remarkably well with the observed trends in all three periods. Calculated trends of B10LP from PDO are more than a factor of five too low for both periods 1957–2005 and 1957–2017, while no significant trend is found for 1983–2009. The calculated trend of B10LP from AMO agrees with the observed value during 1983–2009, but no significant trend is found for the two longer periods 1957–2005 and 1957–2017. Since the trends of longer periods should carry more weight, results in Table 3 suggest that GT is the primary contributor to the linear trends in B10LP, the maximum possible contribution from PDO is about 10%, while contribution from AMO and Niño3.4 is negligible. These results are consistent with the proposal by Liu et al. (2015) that the reduction in light precipitation days in China is part of the extensive worldwide reports of enhancements in heavy precipitation and reductions in the light and moderate precipitation under global warming environment. Finally, it is worth mentioning that the value of $\Delta$B10LP/$\Delta$GT calculated from detrended data is consistent with those shown in Table 3.

In summary of Sec. 3.2, our study suggests that the reduction of cloud cover in China is primarily driven by the MCL-Feedback cycle under global warming environment, PDO plays a secondary role, while the contribution from AMO and Niño3.4 is insignificant. This finding lends strong albeit regional support for the findings in Sec. 3.1, because it is derived primarily via an arithmetic analysis; the only exception is in attributing the cause of reduction trend in the bottom 10% light precipitation to GT, which is adopted from extensive investigations cited in Trenberth et al. (2003) and Liu et al. (2015).

## 4 Summary and conclusions

Worldwide satellite observations (ISCCP, 1983–2009) of linear trends in cloud cover are compared to those in global precipitation (GPCP pentad V2.2 1983–2009), to decipher possible cause(s) of the trends in cloud cover. The spatial distributions of the linear trends of total cloud cover and precipitation are characterized primarily by a broadening of the center of precipitation (ascending/wet zone of Hadley cells) over the Maritime Continent in all directions (Zhou et al., 2011; Liu R. et al., 2016;). The underlying mechanism driving the broadening is proposed to be the moisture–convection–latent heat

feedback cycle under increasing SST conditions (Trenberth et al., 2003). Our correlation studies show that global warming, AMO and PDO can each explain significant spatial variabilities of the linear trends in cloud cover (67%, 49% and 38%, respectively) and precipitation (86%, 59% and 53%, respectively). Contribution by Niño3.4 is insignificant. A linear combination of global warming and AMO can explain as much as 74% and 79%, respectively, of the spatial variabilities of linear trends in cloud cover and precipitation.

Taking advantage of the extensive daily observations of cloud cover and precipitation from Chinese surface meteorological stations over a relatively long period (1957–2005), a quantitative matching relationship between linear trends in cloud cover and precipitation is established via an arithmetic analysis, which is more robust than the correlation method. Furthermore, our study suggests that the reduction of cloud cover in China is also primarily driven by the moisture–convection–latent heat feedback cycle under increasing global temperature conditions (Trenberth et al., 2003), PDO plays a secondary role, while the contribution from AMO and Niño3.4 is insignificant because neither has any linear trend during 1957–2005. This finding lends strong albeit regional support for the findings in Sec. 3.1, because it is derived primarily via an arithmetic analysis.

The long-term radiative effect of aerosols on the global temperature and other climate parameters are expected to be imbedded in the observed changes of these climate parameters, and thus included in this study. The effects of anthropogenic aerosols on clouds and precipitation by acting as cloud condensation nuclei (CCN) is a highly controversial issue, which has been discussed extensively in a number of studies as well as one of our earlier papers (Liu et al., 2015). We defer the discussion on this issue to future studies, and acknowledge here that the aerosol could introduce an unknown degree of uncertainty in this study.

Cautionary statements: It is important to note that many critical analyses in Sec. 3 have utilized some sort of correlation analysis, which does not prove a causal relationship, nor does a higher correlation coefficient imply a more important causal relationship. The attribution of cause–effect can only be established if a mechanistic model that is based on the cause/mechanism, can successfully reproduce the linear trends in cloud cover quantitatively. Until such a model exists, all correlation results should be used only as suggestions or hints of a possible causal relationship. Unfortunately, the task is extremely challenging for current climate models as they tend to have large uncertainties in the simulation of key atmospheric parameters, particularly for clouds and precipitation (Flato et al., 2013). A second caveat is that both the ISCCP and GPCP datasets have utilized IR related data to produce the final products. However, ISCCP comprises merged visible channels and other available channels, while GPCP comprises merged microwave channels and gauge data; in fact, the microwave channels play a more important role than the IR data. In conclusion, these two datasets share relatively limited common data in the IR channels, but both datasets merge substantial independent data sources in the visible and microwave channels. Therefore, we believe that the correlation between cloud and precipitation should not be significantly affected by their common data source.

*Data availability.* Corrected satellite cloud cover data were obtained from NCAR UCAR RDA (https://rda.ucar.edu/datasets/ds741.5/, last access: 10 June 2020). The gridded precipitation data were obtained from NOAA NCDC at ftp.ncdc.noaa.gov/pub/data/gpcp (last access: 10 June 2020). Annual average of global temperature anomaly were

obtained from NCDC (https://www.ncdc.noaa.gov/cag/global/time-series/globe/land_ocean/ann/12/1957-2005, last access: 10 June 2020) and PDO, AMO, Niño3.4 are from NOAA WGSP (http://psl.noaa.gov/gcos_wgsp/Timeseries/, last access: 10 June 2020). Daily cloud cover and precipitation data of Chinese surface meteorological stations were obtained from CMA (http://data.cma.cn/site/index.html). The data of this paper are available upon request to Shaw Chen Liu (shawliu@jnu.edu.cn).

*Author Contributions.* SL proposed the essential research idea. XZ performed the analysis. SL, XZ and RL drafted the manuscript. XW, JM and YL helped analysis and offered valuable comments. All authors have read and agreed to the published version of the manuscript.

*Competing interests.* The authors declare that they have no conflict of interest.

*Acknowledgments.* The authors thank the China Meteorological Data Service Center and National Climatic Data Center for providing datasets that made this work possible. We especially thank J. Norris for providing the corrected ISCCP data. We also acknowledge the support of the Institute for Environmental and Climate Research in Jinan University.

*Financial support.* This research was supported by National Natural Science Foundation of China (grant number 91644222, 41805115), Guangzhou Municipal Science and Technology Project, China (grant number 202002020065), and Fundamental Research Funds for the Central Universities (grant number 21618322).

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

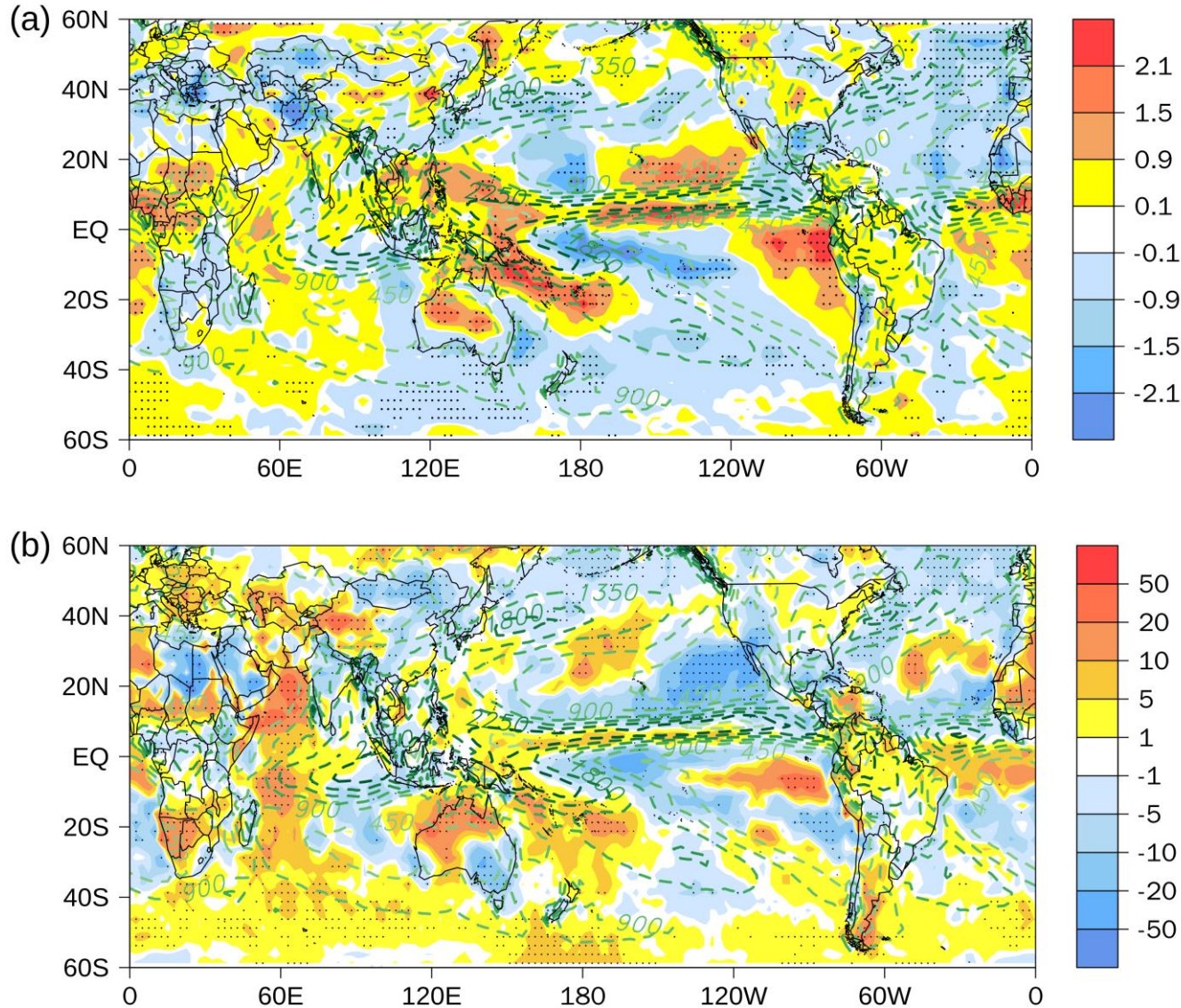

Figure 1. (a) Trends in total cloud cover (units: % per decade) from corrected ISCCP D2 data set (1983–2009). (b) Trends in annual total precipitation (units: % per decade) from GPCP pentad V2.2 (1983–2009). Dots indicate changes significant at the 95% confidence level. Dashed green contours indicates the climatology of total precipitation (units: mm per year).

500

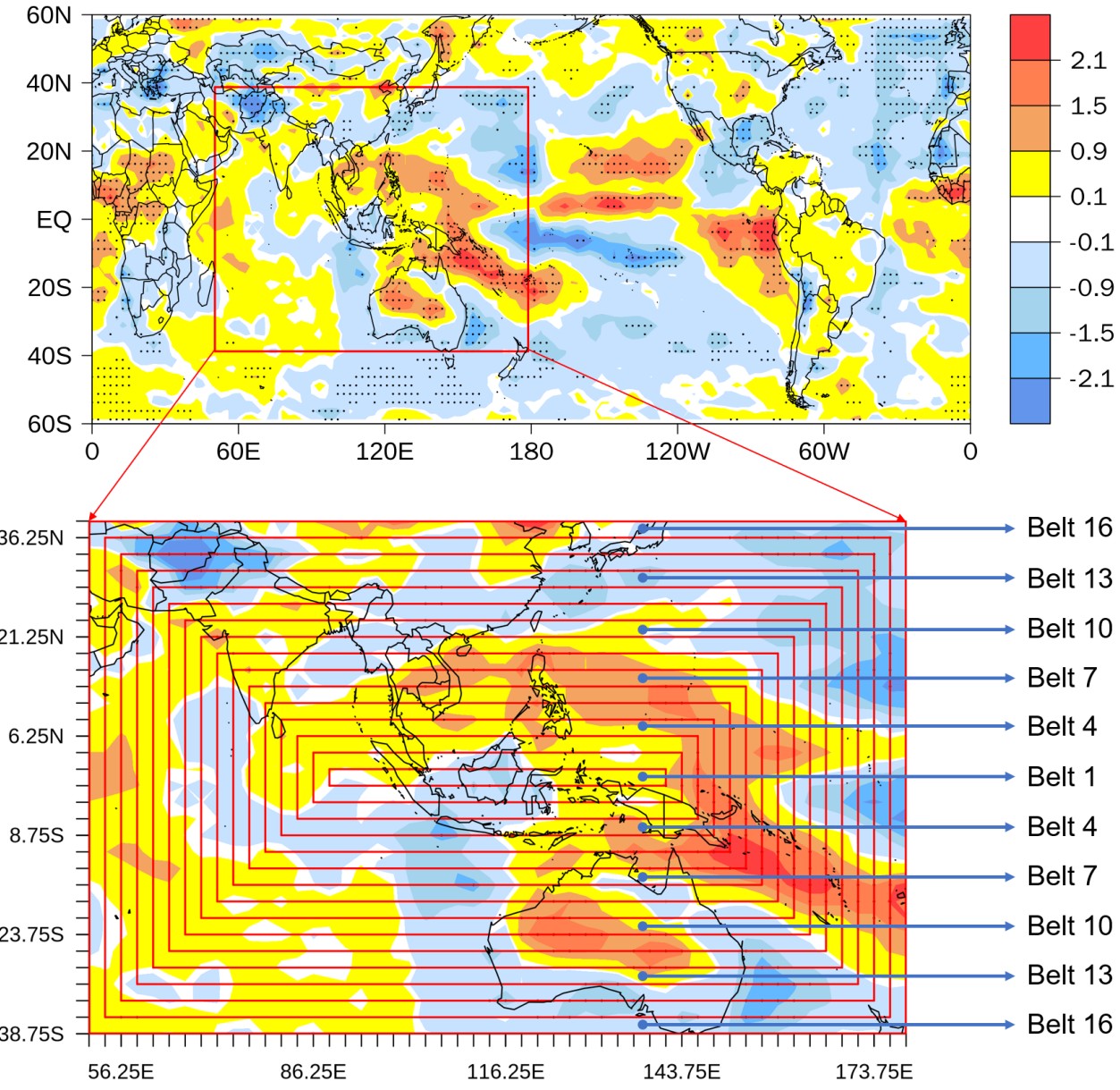

Figure 2. Upper map is reproduced from Fig. 1a. The 16 rectangular belts of 2.5 degree wide in both latitude and longitude centered in the middle of Kalimantan, Indonesia encompass the major ascending/wet zone of Hadley cell. The expansion of cloud cover and precipitation relative to these belts are used as a measure of the broadening of the major ascending zone of Hadley circulation.

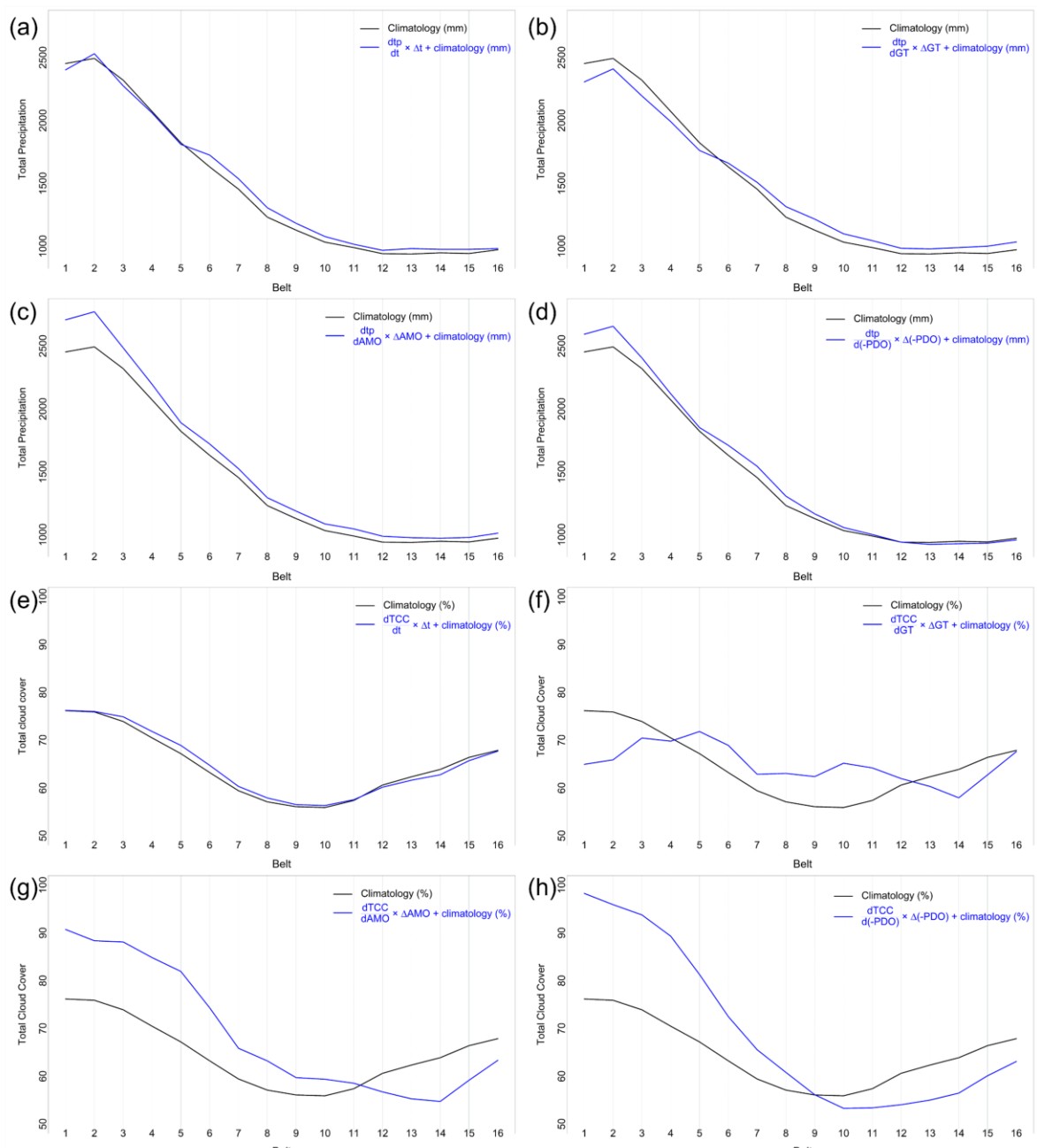

Figure 3. Changes (blue curve) from the climatology (black) during the period 1983–2009 in the annual total precipitation (mm) in the 16 belts of Figure 2 as a function of time (a), global temperature (b), AMO (c) and PDO (d). Changes from the climatology in the annual total cloud cover (%) in the 16 belts of Figure 2 as a function of time (e), global temperature (f), AMO (g) and PDO (h). The formula for calculating the blue curve, for instance for the changes in precipitation as a function of global temperature (Fig. 3b) is d(TP)/d(GT)×ΔGT where ΔGT denotes difference in the global temperature between 1983 and 2009.

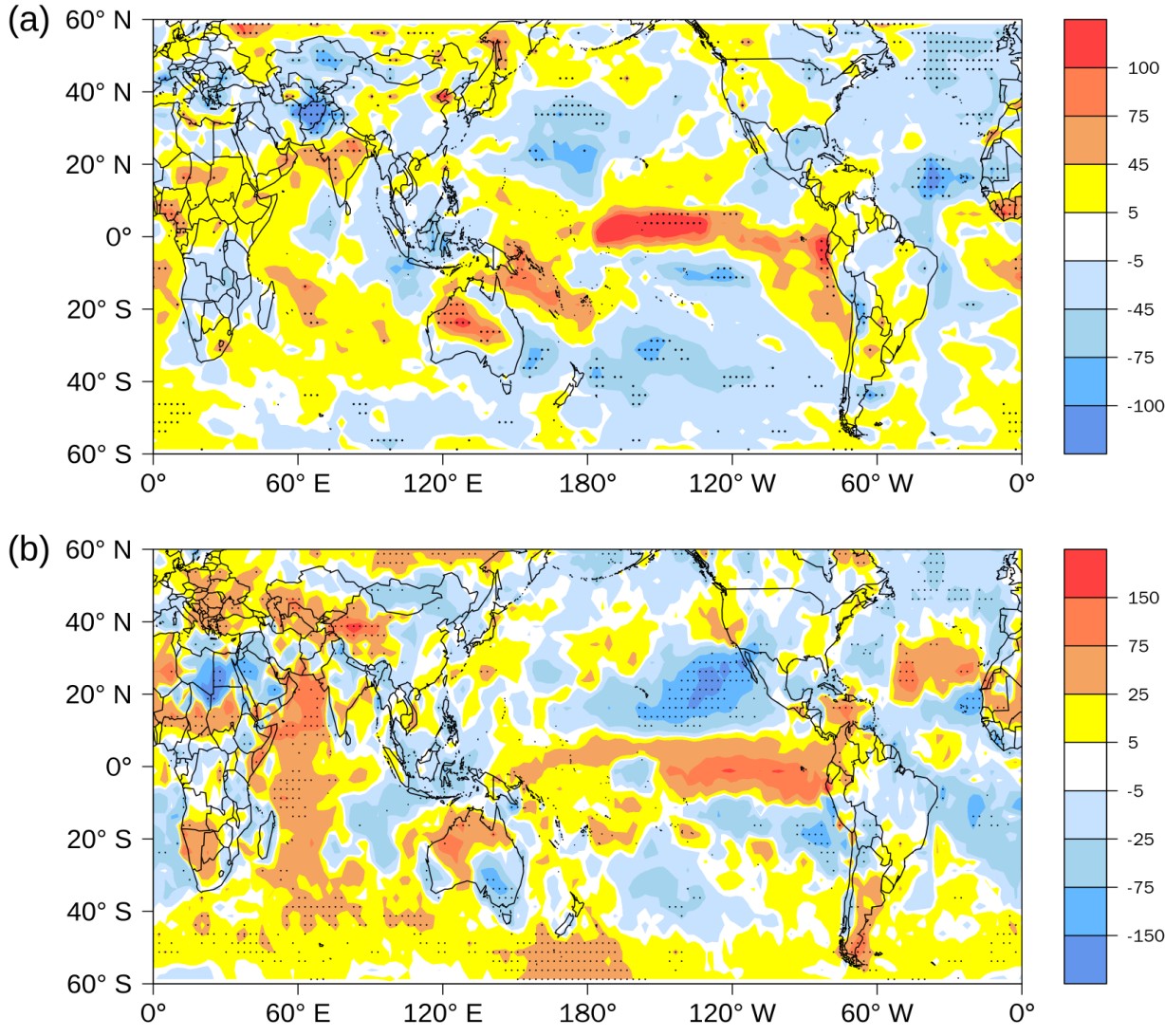

**Figure 4 (a) Slope of linear regression between total cloud cover and global temperature anomalies (units: % K⁻¹) at individual grids from corrected ISCCP D2 data set (1983–2009). (b) Slope of linear regression between annual total precipitation and global temperature anomalies (units: % K⁻¹) at individual grids from GPCP pentad V2.2 (1983–2009). Dots indicate changes significant at the 95% confidence level.**

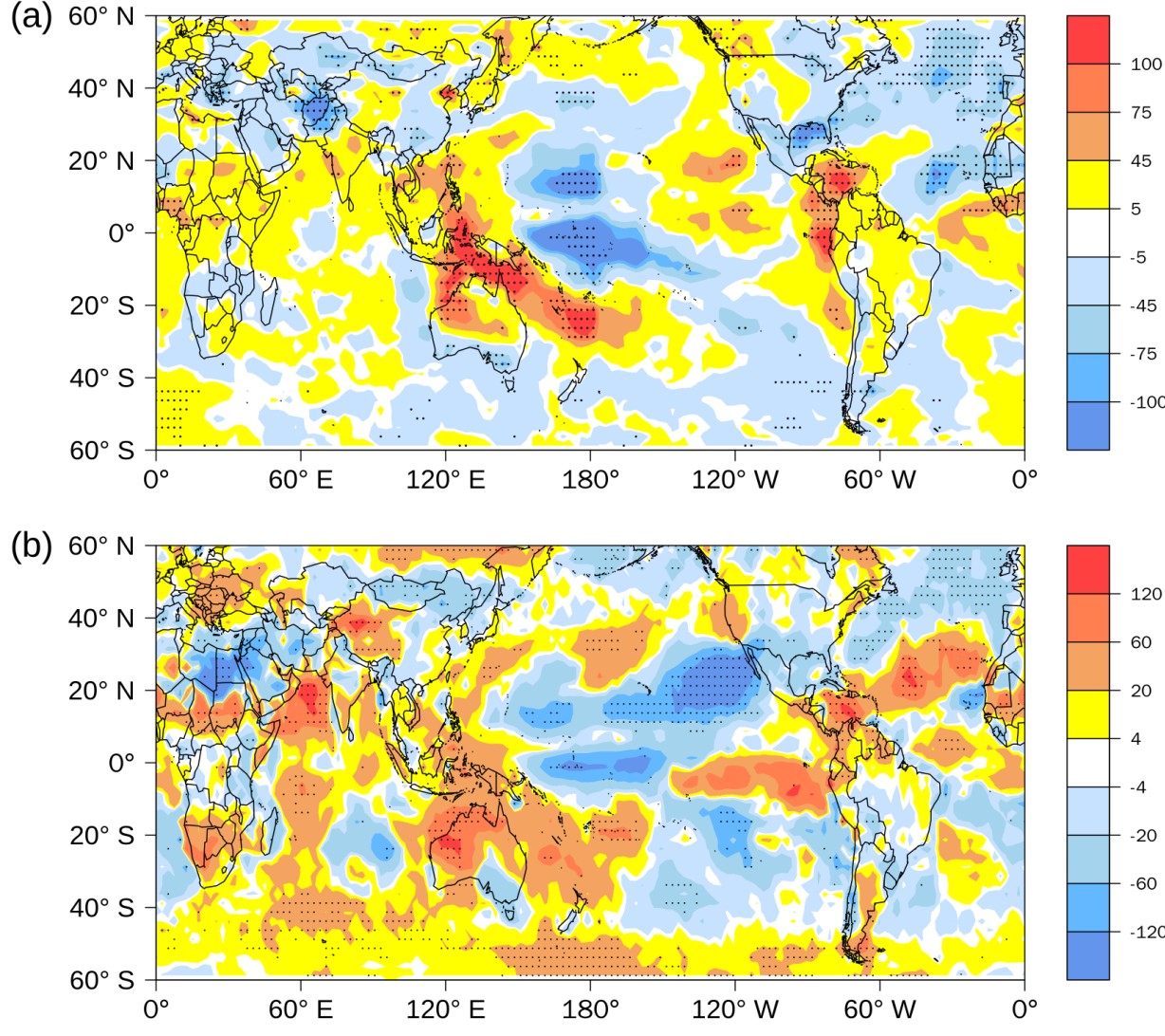

**Figure 5. (a) Slope of linear regression between total cloud cover and AMO (units: % K$^{-1}$) at individual grids from corrected ISCCP D2 data set (1983–2009). (b) Slope of linear regression between annual total precipitation and AMO (units: % K$^{-1}$) at individual grids from GPCP pentad V2.2 (1983–2009). Dots indicate changes significant at the 95% confidence level.**

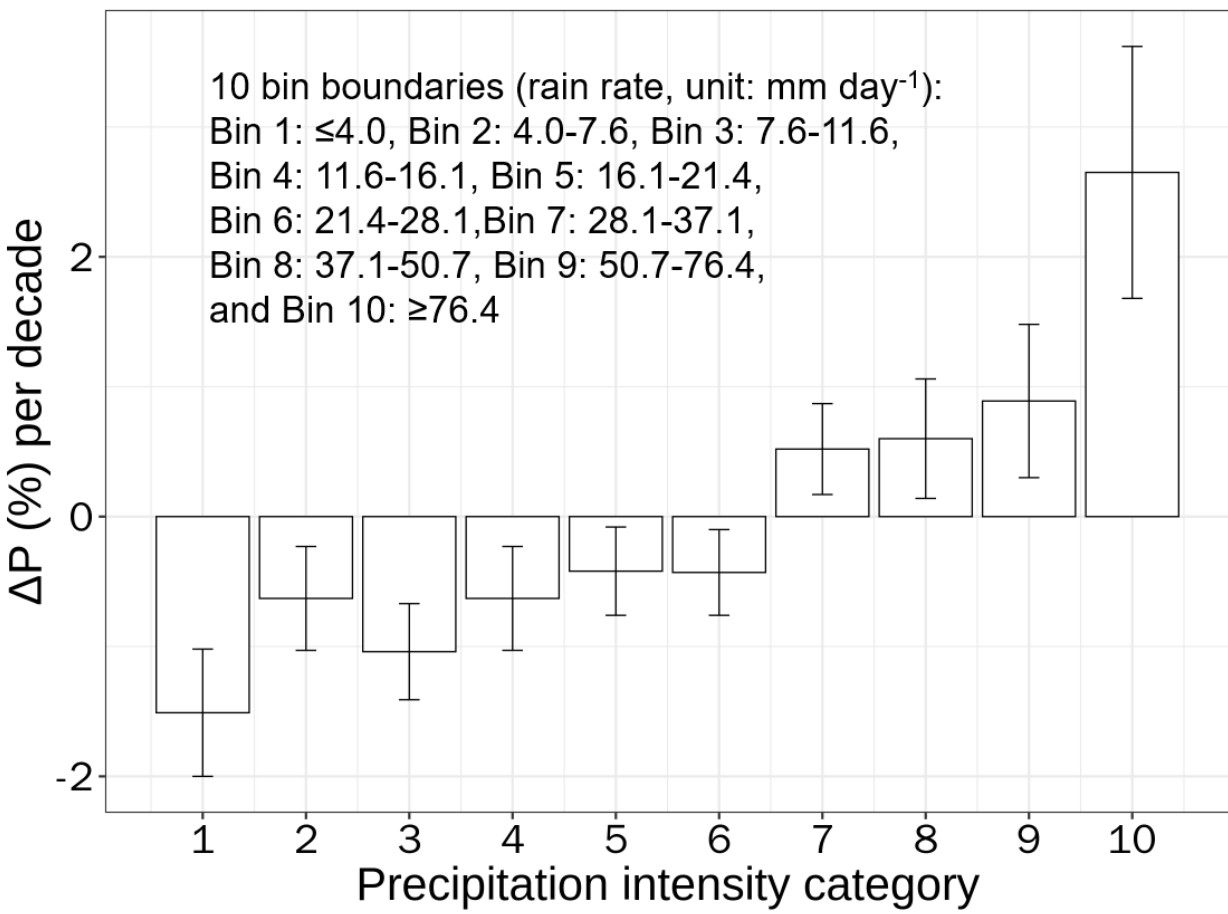

**Figure 6. Linear trends of annual precipitation amount (ΔP) falling within each of the ten intensity bins in China during 1957–2005. The vertical line on top of each bar denotes one standard error.**

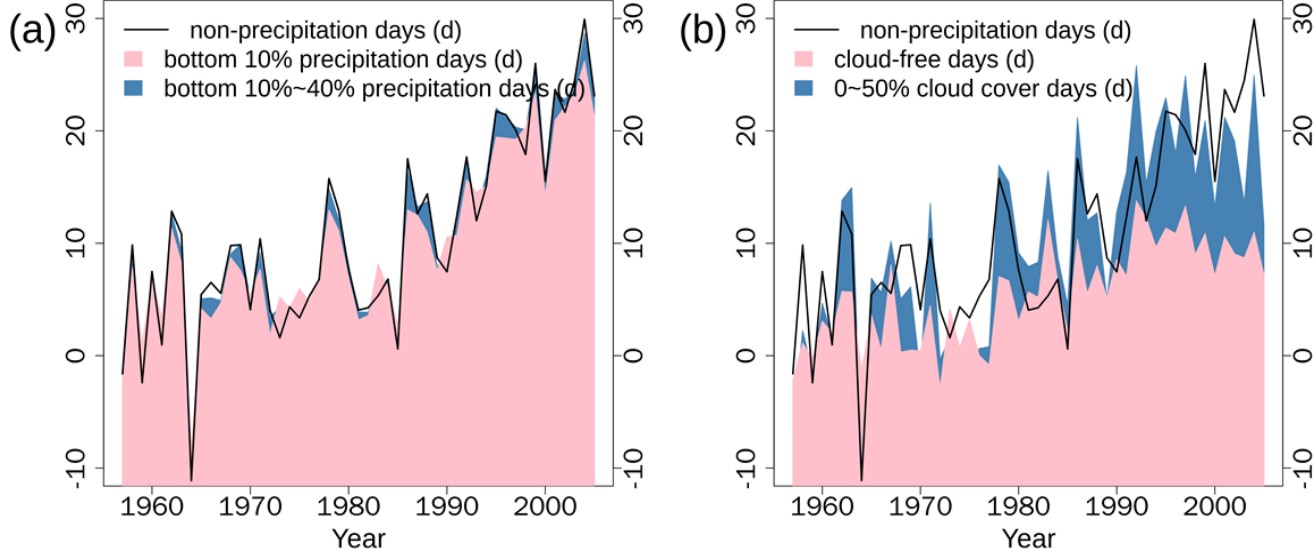

**Figure 7. (a)** Time series of increases in non-precipitation days compared to decreases of bottom 10% and bottom 10~40% precipitation days in 1957–2005. **(b)** As in (a) but for increases in non–precipitation days and cloud–free days compared to decreases in 0–50% cloud cover days. Increases or decreases are calculated as original time series subtract the value of the start year (1957).

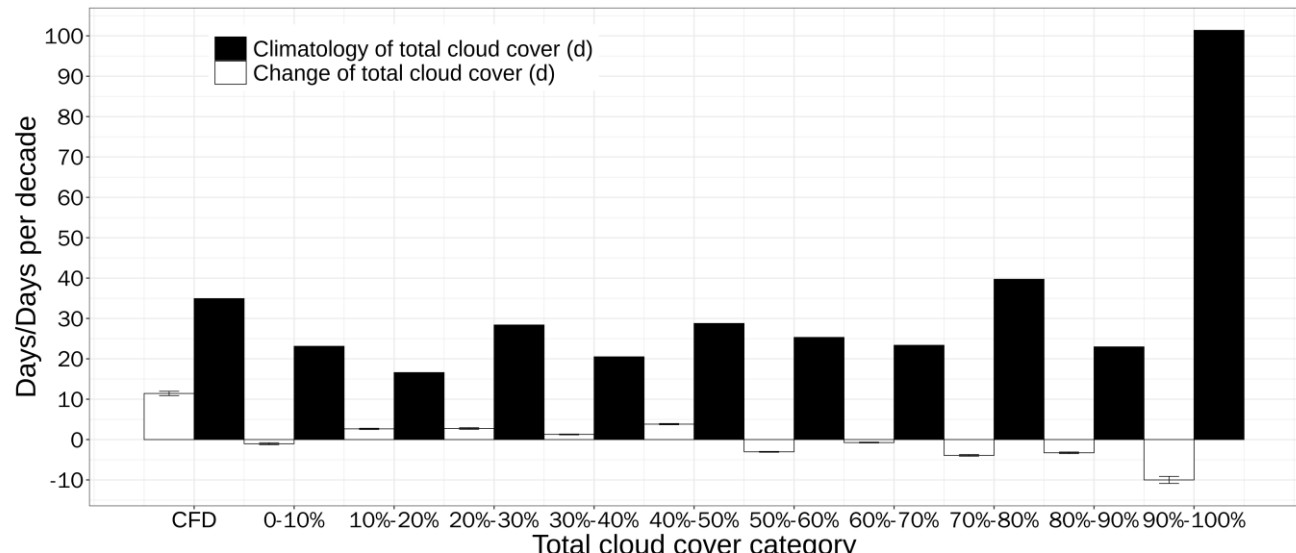

**Figure 8. Climatology (units: days) and changes (units: days per decade) in the cloudy days falling within each bin during 1957–2005. CFD denotes cloud–free days and 0–10% denotes days of cloud cover within the range of (0–10%). The vertical line on top of each bar denotes one standard error.**

**Table 1. Correlation coefficient between spatial distribution of trends of TCC (TP) and those calculated from changes of TCC (TP) as a function of different climatic indexes**

| R | Trend of TCC | Trend of TP |
|---|---|---|
| δ(GT) | 0.82 *** | 0.93 *** |
| δ(-PDO) | 0.62 *** | 0.73 *** |
| δ(AMO) | 0.70 *** | 0.77 *** |
| δ(Niño3.4) | -0.20 *** | 0.02 |
| δ(GT)+δ(-PDO) | 0.74 *** | 0.85 *** |
| δ(GT)+δ(AMO) | 0.86 *** | 0.89 *** |
| δ(GT)+δ(Niño3.4) | 0.89 *** | 0.93 *** |
| δ(-PDO)+δ(AMO) | 0.67 *** | 0.79 *** |
| δ(-PDO)+δ(Niño3.4) | 0.61 *** | 0.72 *** |
| δ(AMO)+δ(Niño3.4) | 0.65 *** | 0.73 *** |
| δ(GT)+δ(-PDO)+δ(AMO) | 0.76 *** | 0.87 *** |
| δ(GT)+δ(-PDO)+δ(Niño3.4) | 0.72 *** | 0.84 *** |
| δ(GT)+δ(AMO)+δ(Niño3.4) | 0.86 *** | 0.88 *** |
| δ(-PDO)+δ(AMO)+δ(Niño3.4) | 0.65 *** | 0.78 *** |
| δ(GT)+δ(-PDO)+δ(AMO)+δ(Niño3.4) | 0.75 *** | 0.86 *** |

Note: GT denotes global temperature anomalies. δ(GT) denotes $\Delta GT \times dTCC/d(GT/GT_\sigma)$ or $\Delta GT \times dTP/d(GT/GT_\sigma)$, where $\Delta GT$ is the change of GT for the studied period and $GT_\sigma$ is the standard deviation of GT, and other factors likewise. *** indicates statistically significant at the 99% confidence level based on student's $t$ test.

**Table 2. Climatology and days changed for precipitation days and cloudy days**

| | Climatology (day) | Change rate (day per decade) | Relative change rate (% per decade) | Change over 49 years (day) | Relative change over 49 years (%) |
|---|---|---|---|---|---|
| NPD | 202.5 | 4.5±0.2 *** | 2.2±0.1 *** | 22.1±1.0 *** | 10.9±0.5 *** |
| B10% | 116.9 | -4.2±0.2 *** | -3.6±0.2 *** | -20.6±1.0 *** | -17.6±1.0 *** |
| B20% | 132.0 | -4.3±0.2 *** | -3.3±0.2 *** | -21.1±1.0 *** | -16.0±1.0 *** |
| B30% | 141.2 | -4.4±0.2 *** | -3.1±0.1 *** | -21.6±1.0 *** | -15.3±0.5 *** |
| B40% | 147.5 | -4.5±0.2 *** | -3.1±0.1 *** | -22.1±1.0 *** | -15.0±0.5 *** |
| T60% | 15.0 | 0±0 | 0±0 | 0±0 | 0±0 |
| CFD | 34.9 | 2.3±0.1 *** | 6.6±0.3 *** | 11.3±0.5 *** | 32.3±1.5 *** |
| ≤50% | 152.3 | 4.3±0.2 *** | 2.8±0.2 *** | 21.1±1.0 *** | 13.7±1.0 *** |
| >50% | 212.7 | -4.3±0.2 *** | -2.0±0.2 *** | -21.1±1.0 *** | -9.9±1.0 *** |

Note: *** indicates statistically significant at the 99% confidence level based on student's $t$ test. NPD denotes non-precipitation days, B10% denotes bottom 10% precipitation days, T60% denotes top 60% precipitation days, ≤50% denotes ≤50% cloud cover days and CFD denotes cloud-free days.

**Table 3 Comparison of observed linear trends of bottom 10% light precipitation with calculated trends for three time periods**

| Unit: % per decade | 1957–2005 | 1957–2017 | 1983–2009 |
|---|---|---|---|
| Observed trend | -1.51±0.49 | -2.02±0.37 | -2.44±1.29 |
| Calculated from GT | -1.81±0.24 | -2.31±0.16 | -2.96±0.70 |
| Calculated from PDO | -0.33±0.09 | -0.21±0.01 | Insignificant |
| Calculated from AMO | Insignificant | Insignificant | -2.45±0.46 |