# Peer review of "Figure S1. Spatial distribution of 477 stations used in this study."

_Atmospheric Chemistry and Physics, 2020_

## Referee Comment (RC1) · Anonymous Referee #3 · 27 Aug 2020

Review of Zhong et al. 2020

The authors present two analyses concerning trends in clouds and rainfall. One uses global, satellite-observed cloud and precipitation data to show that cloud cover and precipitation trends are consistent with an expanding tropical belt. The other looks at surface-observed clouds and rain rates in China to show that light, stratiform rain and overcast clouds are declining while convective rain associated with more broken clouds is relatively more common. These results are consistent with prior work showing a widening tropical belt and a trade-off from stratiform precipitation in favor of convective precipitation.

[Figure]

The work addresses some very large and interesting problems using a fairly simple and easy to understand method, which is commendable. The quality and presentation of the manuscript is high and the work presents great value to the community. There are a few places where the analysis needs a bit more rigor, especially regarding the removal of long-term variation from timeseries in the correlation analysis. It is crucial that we know that the correlations we see are due to interannual variations and not due to coinciding trends. If the authors can do this bit of extra work, the results will be significantly more robust.

Major comments:

There is talk of a widening Hadley cell, and the results do hint at this, but I would love to see a bit more rigor in 1) defining what your data show as the tropical belt, maybe with a zonal mean plot showing the mean clouds/precipitation for latitude zones, then 2) showing the mean trends for the same zones. You could do this globally, or for a specific region between longitude bounds.

I'm not completely convinced by the trend/correlation analysis discussed in Figure 3 and the associated tables. Specifically, I'm concerned that linear trends in timeseries being correlated may occur coincidentally and that this could be driving much of the signal in Figure 3. The authors need to show that the relationships between global temperature and regional variations in cloud cover and precipitation are consistent when the linear trends (or long-term variability with very few independent data points) are removed. This removal could be done either by detrending the time series or by filtering out a 5-year or 10-year running mean. The maps showing significant relationships after this filtering will more clearly show how year-year global temperature variations interact with year-year cloud and precipitation variations. Basically, the idea is that if temperature is actually driving cloud and precip changes, then the relationship should be apparent on both decadal and yearly timescales. To aid in this, you could also show a few time series plots for some significant regions as an example, showing that year-year temperature and cloud variations are similar, most importantly by adding a

temperature plot to Figure 2.

Line 105 & 106: Can you clarify this? It sounds like you mean that you chose stations that have consistent reporting throughout the year. Can you also clarify whether observation timing throughout the diurnal cycle remains consistent for those years? Are you excluding any night data if lunar illumination is insufficient, or can you show that interannual variation of daytime data is equivalent to night?

Minor comments:

Line 49: I think you may be referring to Eastman, Warren, and Hahn (2011) that uses ocean observations. The 2013 paper is only concerned with land stations.

Can you list the grid spacing of all data? The precip data is 2.5x2.5 and it appears that the clouds are at that resolution as well? The spacing itself appears appropriate, with little spurious-looking noise in the contour plots.

I think you need one more sentence describing the Norris and Evan empirical method for removing spurious trends, something like: "by removing anomalous cloud variability within individual grid boxes shown to be associated with artifact factor anomalies", which is (somewhat lazily) adapted from their abstract.

Figure 1: It's frustrating that the contours of total precipitation aren't plotted in the mid-latitude storm tracks, but the trends seem to be plotted in these regions. Can you explain this discrepancy, or better yet, plot the climatological average precipitation in the regions where you plot the trends? There appear to be some regions, especially the N Atlantic where precip contours vanish. The chosen contour interval may not be sensitive enough to show variability in many regions, which is why there aren't contours plotted. Could you tighten the interval for total precip values below 900? This would really aid the paper since the southern ocean storm track and N Atlantic also appear to have a significant precipitation trends.

Figure 5: Can you provide numbers that show what these bins mean? What intensity

[Figure]

of rain occurs in bin 10, for instance? Line 198 says bins are 'equal'. Does this mean equal number of obs per bin, or equal ranges of rain rate within each bin?

---

## Referee Comment (RC2) · Anonymous Referee #2 · 4 Sep 2020

The main focus of this paper is establishing the role of global warming, AMO, and PDO in the spatial pattern of global cloud and precipitation trends (based on global satellite records). Cloud cover and precipitation trends from Chinese meteorological stations are also examined.

Unfortunately, I find a number of major flaws in this paper and do not believe that it meets the quality for publication in ACP at this time:

1) There is a lot of overlap with recent papers that have performed similar analyses, and I struggle to see how this paper provides a substantial new contribution to the peer-reviewed literature. Figure 1a is nearly identical to Figure 1a in Norris et al. (2016),

the PDO/AMO analysis is similar to that in Chen et al. (2019), and Adler et al. (2017) already examine contributions of the PDO and AMO to global precipitation trends.

Adler, R.F., Gu, G., Sapiano, M. et al. Global Precipitation: Means, Variations and Trends During the Satellite Era (1979–2014). Surv Geophys 38, 679–699 (2017). https://doi.org/10.1007/s10712-017-9416-4

2) How reliable are the trends in the satellite data products? While the authors use the corrected data set of Norris and Evan (2015) to account for some of these issues in the ISCCP data, no mention is made of the reliability of the trends in the GPCP precipitation data set (line 91). Also, no discussion is provided of the role that potential instrumentation/reporting method changes may play in the trends from the Chinese meteorological stations.

3) Trends in cloud cover and precipitation are attributed to global warming, AMO, and PDO over the 1983-2009 period, yet this is a very short interval for isolating signatures from decadal modes of variability. Additionally, all three of these indices (global temperature, AMO, and PDO) experience trends over this period. So, is this period even long enough to attempt an analysis like this, because it's less than one full oscillation for the PDO and AMO? How do you have enough degrees of freedom to accurately identify the pattern of cloud and precipitation anomalies associated with the PDO and AMO and distinctly separate it from the global warming trend contribution? And, just because global temperatures are warming, it doesn't mean that concurrent trends in clouds and precipitation are necessarily caused by global warming. The similarity in Figs. 1 and 3 is by construction, as the global temperature time series is dominated by an increasing trend (so any trend in clouds and precipitation will by definition be highly correlated with global temperature). It would be better to define Figure 3 using a detrended global temperature timeseries (as Reviewer #3 also suggests).

Another related concern is a lack of independence of the global temperature, AMO, and PDO indices (because they all have trends over the 1983-2009 interval). How can

the global warming trend explain 67% of the variance in the global cloud cover trends and the AMO trend explain 49% (line 158)? You can't explain more than 100% of the variance, unless the indices are not independent of one another. In other words, it doesn't appear that the global warming, PDO, and AMO indices are actually orthogonal to one another (as is claimed on lines 166-167).

4) The authors are examining cloud and precipitation features in the deep tropics and attributing them to a poleward shift in the Hadley cell edge and midlatitude jet streams (lines 131-132, 138-140). The expansion of the Hadley cell and poleward shift of the jet streams affects precipitation in the subtropics and midlatitudes (poleward of 30 degrees latitude), not in the deep tropics. For tropical precipitation changes, the authors need to really be comparing their results with recent changes in the ascending branch of the Hadley cell (Intertropical Convergence Zone), not the descending branch in the subtropics.

5) Section 3b seems like a separate study and to not be related to the rest of the paper. Trends in a small region are not necessarily affected by global drivers, and regional influences are not discussed at all. This data analysis also suffers from similar problems as the global analyses in section 3a (see major comments #2 and #3).

Minor Revisions

Lines 20-29: The trends described in this paragraph do not appear to closely match those shown in Norris et al. (2016), especially over land and over the Indian Ocean.

Lines 54-71: Somewhere in this paragraph, it is probably worth mentioning that the constraint on global precipitation is 2–3% per K, and not 7% per K. See, for example, Jeevanjee and Romps (2018; https://doi.org/10.1073/pnas.1720683115).

Line 69, 131-132, 138-140: See major comment #4. The expansion of the Hadley cells has nothing to do with enhancement of tropical precipitation. It is related to sub-tropical static stability (Chemke and Polvani 2019: https://doi.org/10.1175/JCLI-D-18-

0330.1). If anything, an expansion of tropical precipitation would contradict the literature, which suggests a narrowing of the Intertropical Convergence Zone in a warming climate (Byrne and Schneider 2016: https://doi.org/10.1002/2016GL070396; Su et al. 2017: https://www.nature.com/articles/ncomms15771).

Line 160: The figure for the PDO really belongs in the main body of the paper, as it is part of the main conclusions of the paper (see abstract).

Line 187: No, the key difference here is that Chen et al. (2019) use the first 300 years of control model simulations to define the cloud cover patterns associated with the PDO and AMO, which avoids the issues of concurrent trends in the indices using the observations (see major comment #3 above).

Lines 189-193: Why is the PDO deemed insignificant here? Is this based entirely on Eastman and Warren's analysis? Nothing shown in this paper appears to make the PDO less significant than the AMO (see Table 1).

Lines 208-210: Could the increase in non-precipitation days and decrease in light precipitation days reflect a change in reporting method? How do you know that these changes are in fact physical?

Lines 237: Difficult to read as written. The equation should be spaced out.

Figures: I would suggest inverting the color bar such that blues correspond to more clouds/precipitation and reds correspond to less.

Table 1: How are you evaluating significance? I have a difficult time believing that a correlation of 0.02 is still significant at the 95% confidence level. Are you taking into account autocorrelations among neighboring grid points, which would greatly reduce the number of degrees of freedom in your t-test?

Table 2: Similarly, how is significance being evaluated here? A trend of 0% (see T60%) should not be statistically significant at all, especially at the 99% level.

Typos

Line 20: are of great importance

Line 27: places affiliated to Australia – not sure what this means, please rephrase

Line 98: provided by

Line 99: retained

Line 105-106: Incomplete sentence . . . please rewrite.

Line 145: is robust

Figure 6a: bottom 10%-40%

---

## Referee Comment (RC3) · Anonymous Referee #1 · 6 Sep 2020

Summary

This is a relatively straightforward paper that reassess changes in both cloud cover and precipitation, and the possible causes of these changes. Which is an important endeavor. Using global satellite data (e.g., corrected ISCCP data and GPCP data), the authors first show similar changes in cloud cover and precipitation, particularly over the Maritime continent, and suggest these changes are largely consistent with widening of the tropical belt (and the moisture-convection-latent heat feedback). They go on to associate a significant percentage of these changes mainly to global warming, but also the AMO. These results are based on correlation/regression analysis alone. In a

somewhat disconnected Part 2 of the paper, the authors focus on China, and investigate clouds and precipitation trends from nearly 500 surface stations over a longer time period. Here, the authors argue the decrease in cloud cover and overall shift toward higher precipitation intensity is due to global warming, and the moisture-convection-latent heat feedback.

Comments

In terms of the indices that are looked at to understand the cloud and precipitation changes, the authors focus on global mean temperature, as well as the PDO, ENSO (Nino3.4 SST) and AMO. However, Norris et al. (2016) also argued for the importance of volcanic aerosol in explaining the cloud changes (as described in the Introduction). To some extent, this volcanic aerosol signal should appear in the global mean surface temperature. Any thoughts on how to disentangle this? Any thoughts on the possible importance of volcanic aerosol, and recovery from their cooling? Or is this not important, based on the authors analysis?

The conclusion that the PDO is not very important to the cloud and precipitation changes (which the authors argue are primarily due to tropical widening) is inconsistent with several studies that have argued the PDO is associated with tropical widening/contraction. For example:

Allen, R., Norris, J. & Kovilakam, M. Influence of anthropogenic aerosols and the Pacific Decadal Oscillation on tropical belt width. Nature Geosci 7, 270–274 (2014). https://doi.org/10.1038/ngeo2091

And more generally, others have argued for the importance of natural variability in driving recent tropical expansion (as opposed to global warming, at least over the relatively short time period considered). For example:

Allen, R. J., and M. Kovilakam, 2017: The Role of Natural Climate Variability in Recent Tropical Expansion. J. Climate, 30, 6329–6350

[Figure]

Mantsis, D. F., Sherwood, S., Allen, R., and Shi, L. (2017), Natural variations of tropical width and recent trends, Geophys. Res. Lett., 44, 3825– 3832,

Grise, K. M., and Coauthors, 2019: Recent Tropical Expansion: Natural Variability or Forced Response?. J. Climate, 32, 1551–1571

Can these points—particularly the prior conclusion related to the importance of natural variability—be commented on and incorporated into the paper?

The conclusion that the cloud and precipitation changes are consistent with tropical widening is a bit "hand-wavy". Can the authors better quantify this, with an actual analysis of the data, in the context of tropical edge displacements?

It is also unclear how the authors associate tropical widening to the moisture-convection-latent heat feedback. This feedback in largely a thermodynamic feedback, related to global warming and CC scaling. And it seems to largely explain why we would expect less light/moderate precipitation, but more heavy precipitation, under warming. So how does it also explain tropical widening? Is dynamics not important here? Several dynamical mechanisms have been proposed.

L179 "Direct effect of anthropogenic aerosols on clouds and precipitation in the tropical zone is expected to be small as the majority of aerosol emissions are at northern hemisphere mid–latitudes." Is this true? Aren't there quite a lot of tropical aerosol emissions, for example biomass burning?

I suggest including the time series of the climate indices used here (perhaps in the Supplement). The AMO that the authors use is said to have the global warming signal removed. It would be nice to see what this looks like (as well as the other indices, e.g., PDO).

Can the authors better connect part 1 (global analysis) and part 2 (China analysis) of this paper? At the least, the authors can add a statement to the abstract that indicates they extend the global analysis by similarly investigating connections between clouds

and precipitation in China, which has a large number of long-running, high-quality surface weather stations, etc. Or something similar, etc.

The abstract also seems to contradict itself. The global analysis largely attributes cloud and precipitation changes to global warming and the AMO. But then the China analysis says the cloud and precipitation changes are largely due to global warming and the PDO, with AMO (and ENSO) playing an insignificant role, consistent with the global analysis. The only thing consistent is the dominance of global warming, right? AMO is important for the global analysis, but is not important for the China analysis.

---

## Author Comment (AC1) · 28 Nov 2020

Dear Editor,

We appreciate the prompt review and would like to thank the three Reviewers' perceptive and helpful comments and suggestions on our manuscript entitled "Observed Trends of Clouds and Precipitation (1983–2009): Implications for Their Cause(s)", Author(s): Xiang Zhong et al., MS No.: acp-2020-577, MS type: Research article. We have carefully considered all comments and suggestions and carried out major revisions as suggested. We believe that the revisions have resulted in a significantly improvement of the paper. Listed below are point-by-point responses to all comments and suggestions of the three reviewers (Reviewer's points in black, our responses in blue).

**Anonymous Referee #1**

**Summary**

This is a relatively straightforward paper that reassess changes in both cloud cover and precipitation, and the possible causes of these changes. Which is an important endeavor. Using global satellite data (e.g., corrected ISCCP data and GPCP data), the authors first show similar changes in cloud cover and precipitation, particularly over the Maritime continent, and suggest these changes are largely consistent with widening of the tropical belt (and the moisture-convection-latent heat feedback). They go on to associate a significant percentage of these changes mainly to global warming, but also the AMO. These results are based on correlation/regression analysis alone. In a C1 ACPD Interactive comment Printer-friendly version Discussion paper somewhat disconnected Part 2 of the paper, the authors focus on China, and investigate clouds and precipitation trends from nearly 500 surface stations over a longer time period. Here, the authors argue the decrease in cloud cover and overall shift toward higher precipitation intensity is due to global warming, and the moisture-convection latent heat feedback.

**Comments**

In terms of the indices that are looked at to understand the cloud and precipitation changes, the authors focus on global mean temperature, as well as the PDO, ENSO

(Nino3.4 SST) and AMO. However, Norris et al. (2016) also argued for the importance of volcanic aerosol in explaining the cloud changes (as described in the Introduction). To some extent, this volcanic aerosol signal should appear in the global mean surface temperature. Any thoughts on how to disentangle this? Any thoughts on the possible importance of volcanic aerosol, and recovery from their cooling? Or is this not important, based on the authors analysis?

This is a very perceptive point. In our deliberation of potential contributors to the cloud and precipitation changes, we have been concentrating on the familiar large-scale climate oscillations but seemingly overlooked relatively short period or regional climatic forcing such as the volcanic aerosol signal of Pinatubo in 1992–1993. It can be seen in a newly added Figure S4 in the Supplement, the Pinatubo signal shows a clear depression in the global temperature of about 0.2 degree in 1992–1993 and recovery in 1994–1995. So the Pinatubo aerosol signal is imbedded in the global temperature change. In regard to how to disentangle this volcanic signal, we believe it would be a great topic for a future study.

The conclusion that the PDO is not very important to the cloud and precipitation changes (which the authors argue are primarily due to tropical widening) is inconsistent with several studies that have argued the PDO is associated with tropical widening/contraction. For example:

Allen, R., Norris, J. & Kovilakam, M. Influence of anthropogenic aerosols and the Pacific Decadal Oscillation on tropical belt width. Nature Geosci 7, 270–274 (2014). https://doi.org/10.1038/ngeo2091

And more generally, others have argued for the importance of natural variability in driving recent tropical expansion (as opposed to global warming, at least over the relatively short time period considered). For example:

Allen, R. J., and M. Kovilakam, 2017: The Role of Natural Climate Variability in Recent Tropical Expansion. J. Climate, 30, 6329–6350 C2 ACPD Interactive comment Printer-friendly version Discussion paper

Mantsis, D. F., Sherwood, S., Allen, R., and Shi, L. (2017), Natural variations of tropical

width and recent trends, Geophys. Res. Lett., 44, 3825– 3832, Grise, K. M., and Coauthors, 2019: Recent Tropical Expansion: Natural Variability or Forced Response?. J. Climate, 32, 1551–1571 Can these points, particularly the prior conclusion related to the importance of natural variability, be commented on and incorporated into the paper? ˘ The conclusion that the cloud and precipitation changes are consistent with tropical widening is a bit "hand-wavy". Can the authors better quantify this, with an actual analysis of the data, in the context of tropical edge displacements?

We appreciate this important comment which was also raised above by Referee#3. In our response to Referee#3 (please see the response with Figures 1 and 2 above), we now have revised the manuscript by adding a quantitative evaluation of the primary tropical widening over the Maritime Continent.

Regarding the importance of PDO, shown in Figures 2b-2d above are the changes (blue curve) from the climatology (1983–2009) (black curve) in the annual total precipitation (mm) of the 16 belts of Figure 1 as a function of global temperature (GT), AMO and PDO, respectively. The formula for calculating the blue curve, for instance for the changes in precipitation as a function of global temperature (Figure 2b), is $d(TP)/d(GT) \times \Delta GT$, where $\Delta GT$ denotes difference in the global temperature between 1983 and 2009.

It can be seen that Figure 2b (GT) agrees very well with Figure 2a both qualitatively and quantitatively; while Figures 2c and 2d have significantly greater positive values (significant widening) compared to the small negative values (contraction) of Figure 2a for the inner 5 belts, resulting in a significant enhancement of the overall precipitation. This discrepancy is crucial, as the global total annual precipitation, which is equal to global evaporation and determined by the global surface energy budget, increases with global temperature at a rather small rate of about 2%–3% $K^{-1}$ (Cubasch et al., 2001). Therefore, based on the results of Figures 2a-2d, we propose that the trend in global temperature, rather than that of AMO and PDO, is the primary contributor to the observed linear trend of precipitation in 1983–2009. Similarly, it can be seen that Figure 2f agrees with Figure 2e significantly better than Figures 2g and 2h, such that the trend

in global temperature, rather than that of AMO and PDO, can be proposed to be the primary contributor to the observed linear trend of total cloud cover in 1983–2009.

It is also unclear how the authors associate tropical widening to the moisture-convection-latent heat feedback. This feedback in largely a thermodynamic feedback, related to global warming and CC scaling. And it seems to largely explain why we would expect less light/moderate precipitation, but more heavy precipitation, under warming. So how does it also explain tropical widening? Is dynamics not important here? Several dynamical mechanisms have been proposed.

Trenberth et al. (2003) summarized the global warming hypothesis by explaining that the precipitation intensity of storms should increase at about the same rate as atmospheric moisture, which is about 7% $K^{-1}$ according to the Clausius–Clapeyron equation. The precipitation intensity could even exceed the 7% $K^{-1}$ because additional latent heat released from the increased water vapour could invigorate the storm and pull in more moisture from the boundary layer, forming a positive feedback cycle (i.e. the moisture-convection-latent heat feedback cycle) and leaving less moisture available for light and moderate precipitation. A comparison of Figure 1 below with Figure 2e above (in our response to referee 3) reveals that the enhancements in precipitation in the tropics (Figure 1) are the major contributor to the tropical widening in observed precipitation (Figure 2e). Since it has been shown by Liu et al. (2016) that the enhancements in precipitation in the tropics are nearly entirely driven heavy precipitation (strong convections), we propose that the tropical widening is primarily driven by the moisture-convection-latent heat feedback.

[Figure]

Figure 1. Trends in annual total precipitation (units: % per decade) from GPCP pentad V2.2 (1983–2009). Dots indicate changes significant at the 95% confidence level. Contours indicates the climatology of total precipitation (units: mm per year).

L179 "Direct effect of anthropogenic aerosols on clouds and precipitation in the tropical zone is expected to be small as the majority of aerosol emissions are at northern hemisphere mid–latitudes." Is this true? Aren't there quite a lot of tropical aerosol emissions, for example biomass burning? I suggest including the time series of the climate indices used here (perhaps in the Supplement). The AMO that the authors use is said to have the global warming signal removed. It would be nice to see what this looks like (as well as the other indices, e.g., PDO).

Excellent point, we have included the time series of the climate indices used in the Supplement (Figure S4). We also have replaced the remark of "Direct effect of anthropogenic aerosols on clouds and precipitation…" with "Direct effects of anthropogenic aerosols on clouds and precipitation tend to be regional and/or sub-yearly time scale, which are beyond the scope of discussion in this study."

Can the authors better connect part 1 (global analysis) and part 2 (China analysis) of this paper? At the least, the authors can add a statement to the abstract that indicates they extend the global analysis by similarly investigating connections between clouds and precipitation in China, which has a large number of long-running, high-quality surface weather stations, etc. Or something similar, etc. The abstract also seems to contradict itself. The global analysis largely attributes cloud and precipitation changes

to global warming and the AMO. But then the China analysis says the cloud and precipitation changes are largely due to global warming and the PDO, with AMO (and ENSO) playing an insignificant role, consistent with the global analysis. The only thing consistent is the dominance of global warming, right? AMO is important for the global analysis, but is not important for the China analysis.

Thanks for a very thoughtful and helpful comment! We have significantly revised the abstract to better connect part 1 (global analysis) and part 2 (China analysis) of this paper, and to address consistency between part 1 and part 2, as shown below.

Further analysis of the widening of the Hadley and Walker circulations (Figures 3a-3h) shows that the trend in global temperature, rather than that of AMO and PDO, is the primary contributor to the observed linear trends of total cloud cover and precipitation in 1983–2009. The underlying mechanism driving this widening is proposed to be the moisture–convection–latent heat feedback cycle under global temperature conditions. The global analysis is extended by investigating connections between clouds and precipitation in China, which has a large number of long-running, high-quality surface weather stations in 1957–2005, which reveals a quantitative matching relationship between the reduction in light precipitation and the reduction of total cloud cover. Furthermore, our study suggests that the reduction of cloud cover in China is primarily driven by the global temperature conditions, PDO plays a secondary role, while the contribution from AMO and Niño3.4 is insignificant, consistent with the global analysis.

---

## Author Comment (AC2) · 28 Nov 2020

Dear Editor,

We appreciate the prompt review and would like to thank the three Reviewers' perceptive and helpful comments and suggestions on our manuscript entitled "Observed Trends of Clouds and Precipitation (1983–2009): Implications for Their Cause(s)", Author(s): Xiang Zhong et al., MS No.: acp-2020-577, MS type: Research article. We have carefully considered all comments and suggestions and carried out major revisions as suggested. We believe that the revisions have resulted in a significantly improvement of the paper. Listed below are point-by-point responses to all comments and suggestions of the three reviewers (Reviewer's points in black, our responses in blue).

**Anonymous Referee #3 interactive comment**

The authors present two analyses concerning trends in clouds and rainfall. One uses global, satellite-observed cloud and precipitation data to show that cloud cover and precipitation trends are consistent with an expanding tropical belt. The other looks at surface-observed clouds and rain rates in China to show that light, stratiform rain and overcast clouds are declining while convective rain associated with more broken clouds is relatively more common. These results are consistent with prior work showing a widening tropical belt and a trade-off from stratiform precipitation in favor of convective precipitation.

The work addresses some very large and interesting problems using a fairly simple and easy to understand method, which is commendable. The quality and presentation of the manuscript is high and the work presents great value to the community. There are a few places where the analysis needs a bit more rigor, especially regarding the removal of long-term variation from timeseries in the correlation analysis. It is crucial that we know that the correlations we see are due to interannual variations and not due to coinciding trends. If the authors can do this bit of extra work, the results will be significantly more robust.

We appreciate very much for these encouraging comments. As shown below, we have made extensive revisions in point-by-point responses to your comments and

suggestions.

**Major comments:**

There is talk of a widening Hadley cell, and the results do hint at this, but I would love to see a bit more rigor in 1) defining what your data show as the tropical belt, maybe with a zonal mean plot showing the mean clouds/precipitation for latitude zones, then 2) showing the mean trends for the same zones. You could do this globally, or for a specific region between longitude bounds.

We gratefully accept this suggestion by explicitly evaluating the widening of Hadley cell in the observed trends of precipitation and cloud cover "for a specific region between longitude bounds". The results reveal a pleasant surprise, as Figure 2 below provides adequate evidence to show that the trend of global temperature, rather than the trends of AMO and PDO, is the primary contributor to the observed linear trend of precipitation in 1983–2009.

[revised manuscript text omitted]

I'm not completely convinced by the trend/correlation analysis discussed in Figure 3 and the associated tables. Specifically, I'm concerned that linear trends in timeseries being correlated may occur coincidentally and that this could be driving much of the signal in Figure 3. The authors need to show that the relationships between global temperature and regional variations in cloud cover and precipitation are consistent when the linear trends (or long-term variability with very few independent data points) are removed. This removal could be done either by detrending the time series or by filtering out a 5-year or 10-year running mean. The maps showing significant relationships after this filtering will more clearly show how year-year global temperature variations interact with year-year cloud and precipitation variations. Basically, the idea is that if temperature is actually driving cloud and precip changes, then the relationship should be apparent on both decadal and yearly timescales. To aid in this, you could also show a few time series plots for some significant regions as an example, showing that year-year temperature and cloud variations are similar, most importantly by adding a temperature plot to Figure 2.

We agree with you on "linear trends in timeseries being correlated may occur coincidentally and that this could be driving much of the signal in Figure 3". Following your suggestion, we have re-evaluated Table 1 using detrended data of TCC, TP, GT, AMO, PDO and Niño3.4 (Table S1). The correlation coefficients are all less than 0.33, implying that consecutive yearly variabilities contribute insignificantly to the high correlation coefficients in Table 1, and the high correlation coefficients are nearly entirely contributed by the long-term linear trends of GT on PDO and AMO. One of the reasons for the lack of correlation could be due to the small consecutive yearly

variabilities relative to the long-term linear trends (about 0.1) for GT on PDO and AMO (Figure S4).

Table S1 Correlation coefficients of detrended data

| R | Trend of TCC | Trend of TP |
|---|---|---|
| δ(GT) | -0.23 *** | -0.16 *** |
| δ(-PDO) | 0.33 *** | 0.10 *** |
| δ(AMO) | -0.02 | -0.16 *** |
| δ(Niño3.4) | -0.19 *** | 0.05 *** |
| δ(GT)+δ(-PDO) | 0.32 *** | 0.04 *** |
| δ(GT)+δ(AMO) | -0.21 *** | -0.18 *** |
| δ(GT)+δ(Niño3.4) | -0.22 *** | -0.17 *** |
| δ(-PDO)+δ(AMO) | 0.30 *** | 0.04 *** |
| δ(-PDO)+δ(Niño3.4) | 0.32 *** | 0.09 *** |
| δ(AMO)+δ(Niño3.4) | 0.03 ** | -0.15 *** |
| δ(GT)+δ(-PDO)+δ(AMO) | 0.29 *** | -0.01 |
| δ(GT)+δ(-PDO)+δ(Niño3.4) | 0.32 *** | 0.04 *** |
| δ(GT)+δ(AMO)+δ(Niño3.4) | -0.18 *** | -0.18 *** |
| δ(-PDO)+δ(AMO)+δ(Niño3.4) | 0.29 *** | 0.04 *** |
| δ(GT)+δ(-PDO)+δ(AMO)+δ(Niño3.4) | 0.28 *** | -0.01 |

By detrended time series, it was calculated as d(detrended TCC)/(d(detrended GT)/std(detrended GT)); linear trends are the same with the original one.

Line 105 & 106: Can you clarify this? It sounds like you mean that you chose stations that have consistent reporting throughout the year. Can you also clarify whether observation timing throughout the diurnal cycle remains consistent for those years? Are you excluding any night data if lunar illumination is insufficient, or can you show that interannual variation of daytime data is equivalent to night?

The station data used in this study are daily data. The original data we have started calculating were already in the daily time resolution. According to the introduction of this data, the daily data were averaged from four-time measurements (02:00, 08:00, 14:00, 20:00, all in local Beijing time) for each day. Therefore, the night data should be involved in this daily data.

To avoid any more concern about how night data could affect our result, here we cite Kaiser (1998) who also analyzed station data as a proof that daytime data and night data share a similar change for cloud cover (as shown in Figures 3 & 4 below).

[Figure]

Figure 3. Trends in annual mean midday cloud amount for 1951-1994 (percent sky cover per decade). Station trend indicators with circles around them are significant at the 95% confidence level, as are regional trend values that are in bold italics (Kaiser, 1998).

[Figure]

Figure 4. As in Figure R1, but for annual mean midnight cloud amount, 1954-1994 (Kaiser, 1998).

**Minor comments:**

Line 49: I think you may be referring to Eastman, Warren, and Hahn (2011) that uses

ocean observations. The 2013 paper is only concerned with land stations.

Yes, we have added the reference to Eastman, Warren, and Hahn (2011).

Can you list the grid spacing of all data? The precip data is 2.5x2.5 and it appears that the clouds are at that resolution as well? The spacing itself appears appropriate, with little spurious-looking noise in the contour plots.

Done as suggested.

I think you need one more sentence describing the Norris and Evan empirical method for removing spurious trends, something like: "by removing anomalous cloud variability within individual grid boxes shown to be associated with artifact factor anomalies", which is (somewhat lazily) adapted from their abstract.

Done as suggested.

Figure 1: It's frustrating that the contours of total precipitation aren't plotted in the midlatitude storm tracks, but the trends seem to be plotted in these regions. Can you explain this discrepancy, or better yet, plot the climatological average precipitation in the regions where you plot the trends? There appear to be some regions, especially the N Atlantic where precip contours vanish. The chosen contour interval may not be sensitive enough to show variability in many regions, which is why there aren't contours plotted. Could you tighten the interval for total precip values below 900? This would really aid the paper since the southern ocean storm track and N Atlantic also appear to have a significant precipitation trends.

Thanks, we have added more contours.

Figure 5: Can you provide numbers that show what these bins mean? What intensity of rain occurs in bin 10, for instance? Line 198 says bins are 'equal'. Does this mean equal number of obs per bin, or equal ranges of rain rate within each bin?

We have now listed the intensity range of each bin. For Line 198, the phrase has been

changed to "the ten bins of equal rain rate".

---

## Author Comment (AC3) · 28 Nov 2020

Dear Editor,

We appreciate the prompt review and would like to thank the three Reviewers' perceptive and helpful comments and suggestions on our manuscript entitled "Observed Trends of Clouds and Precipitation (1983–2009): Implications for Their Cause(s)", Author(s): Xiang Zhong et al., MS No.: acp-2020-577, MS type: Research article. We have carefully considered all comments and suggestions and carried out major revisions as suggested. We believe that the revisions have resulted in a significantly improvement of the paper. Listed below are point-by-point responses to all comments and suggestions of the three reviewers (Reviewer's points in black, our responses in blue).

**Anonymous Referee #2**

The main focus of this paper is establishing the role of global warming, AMO, and PDO in the spatial pattern of global cloud and precipitation trends (based on global satellite records). Cloud cover and precipitation trends from Chinese meteorological stations are also examined. Unfortunately, I find a number of major flaws in this paper and do not believe that it meets the quality for publication in ACP at this time: 1) There is a lot of overlap with recent papers that have performed similar analyses, and I struggle to see how this paper provides a substantial new contribution to the peer reviewed literature. Figure 1a is nearly identical to Figure 1a in Norris et al. (2016), the PDO/AMO analysis is similar to that in Chen et al. (2019), and Adler et al. (2017) already examine contributions of the PDO and AMO to global precipitation trends. Adler, R.F., Gu, G., Sapiano, M. et al. Global Precipitation: Means, Variations and Trends During the Satellite Era (1979–2014). Surv Geophys 38, 679–699 (2017). https://doi.org/10.1007/s10712-017-9416-4 2)

We agree with the criticism that there are already numerous studies on our subject of study. However, as stated in our introduction, there is hardly any agreement on the quantitative roles of global warming, AMO, and PDO in the spatial pattern of global cloud and precipitation trends. Moreover, there are very few studies utilizing both cloud

and precipitation data sets. Last but not the least, with a lot of help of all three referees' comments, we believe that in the revised manuscript we have made "a substantial new contribution" in the conclusion below: Further analysis of the widening of the Hadley and Walker circulations (Figures 2a-2h, see Response to referee 3) shows that the trend in global temperature, rather than those of AMO and PDO, is the primary contributor to the observed linear trends of total cloud cover and precipitation in 1983–2009. The underlying mechanism driving this widening is proposed to be the moisture–convection–latent heat feedback cycle under global temperature conditions.

How reliable are the trends in the satellite data products? While the authors use the corrected data set of Norris and Evan (2015) to account for some of these issues in the ISCCP data, no mention is made of the reliability of the trends in the GPCP precipitation data set (line 91). Also, no discussion is provided of the role that potential instrumentation/reporting method changes may play in the trends from the Chinese meteorological stations.

This point is well taken. In our study, the reliability of data products is mainly concerned with the precision rather than the absolute accuracy of the data. So comparison of different instruments are usually used to evaluate the reliability of the trends in the ISCCP data or GPCP precipitation data set. For example, Xie et al. (2003) found that good agreement is observed between the pentad GPCP and the gauge-based dataset of Shi et al. (2001) over the combined space–time domain. The correlation is 0.776, 0.660, and 0.688, respectively, for the total value, anomaly, and intraseasonal components of the pentad precipitation. These results imply the reliability of the GPCP pentad data is on the order of 70%, or uncertainty of 30%. For the ISCCP data set Norris and Even (2015) found that the root-mean-square difference between ISCCP and PATMOS-x grid box trends decreases from 2.0% (the amount per decade for the original data) to 0.9% (the amount per decade for the fully corrected data). Disagreement between ISCCP and PATMOS-x cloud trends may be due to differing satellite instruments and methods of cloud retrieval or remaining artifacts in the datasets.

We have made extensive comparisons of the ISCCP data and the GPCP precipitation data with corresponding data at the surface stations in China. In many cases, correlations of better than 0.7 were observed, particularly for precipitation data. Therefore, we believe that the correlation results of 0.7 or better are reliable in this study.

Shi, W., R. W. Higgins, E. Yarosh, and V. E. Kousky, cited 2001: The annual cycle and variability of precipitation in Brazil. NCEP/Climate Prediction Center Atlas, No. 9, National Oceanic and Atmospheric Administration. National Weather Service. [Available online at http://www.cpc.noaa.gov/researchppapers/ncep_cpc_atlas/9/index.html.]

3) Trends in cloud cover and precipitation are attributed to global warming, AMO, and PDO over the 1983-2009 period, yet this is a very short interval for isolating signatures from decadal modes of variability. Additionally, all three of these indices (global temperature, AMO, and PDO) experience trends over this period. So, is this period even long enough to attempt an analysis like this, because it's less than one full oscillation for the PDO and AMO? How do you have enough degrees of freedom to accurately identify the pattern of cloud and precipitation anomalies associated with the PDO and AMO and distinctly separate it from the global warming trend contribution? And, just because global temperatures are warming, it doesn't mean that concurrent trends in clouds and precipitation are necessarily caused by global warming.

Thank you for a highly significant criticism. From a different angle, the other two referees have raised the same concerns. In our response to referee 3, we now have revised the manuscript by adding a quantitative evaluation of the primary tropical widening over the Maritime Continent. Shown in Figures 2b-2d in our response to referee 3 are the changes (blue curve) from the climatology (1983–2009) (black curve) in the annual total precipitation of the 16 belts of Figure 1 (response to referee 3) as a function of global temperature (GT), AMO and PDO, respectively. The formula for calculating the blue curve, for instance for the change in precipitation as a function of global temperature (Figure 2b), is d(TP)/d(GT)*ΔGT, where ΔGT denotes difference

in the global temperature between 1983 and 2009. It can be seen that Figure 2b (GT) agrees very well with Figure 2a both qualitatively and quantitatively; while Figures 2c and 2d have significantly greater positive values (significant widening) compared to the small negative values (contraction) of Figure 2a for the inner 5 belts, resulting in a significant enhancement of the overall precipitation. This discrepancy is crucial, as the global total annual precipitation, which is equal to global evaporation and determined by the global surface energy budget, increases with global temperature at a rather small rate of about 2%–3% $K^{-1}$ (Cubasch et al., 2001). Therefore, based on the results of Figs. 3a-3d, we propose that the trend in global temperature, rather than those of AMO and PDO, is the primary contributor to the observed linear trend of precipitation in 1983–2009. Similarly, it can be seen that Figure 2f agrees with Figure 2e significantly better than Figures 2g and 2h, such that the trend in global temperature, rather than those of AMO and PDO, can be proposed to be the primary contributor to the observed linear trend of total cloud cover in 1983–2009.

The similarity in Figs. 1 and 3 is by construction, as the global temperature time series is dominated by an increasing trend (so any trend in clouds and precipitation will by definition be highly correlated with global temperature). It would be better to define Figure 3 using a detrended global temperature timeseries (as Reviewer #3 also suggests). Another related concern is a lack of independence of the global temperature, AMO, and PDO indices (because they all have trends over the 1983-2009 interval).

Thanks, you are right! In our response to the same question by Referee#3, we have re-evaluated Table 1 using detrended data of TCC, TP, GT, AMO, PDO and Niño3.4 (Table S1 in response to referee 3). The correlation coefficients are all less than 0.33, implying that consecutive yearly variabilities contribute insignificantly to the high correlation coefficients in Table 1, and the high correlation coefficients are nearly entirely contributed by the long-term linear trends of GT on PDO and AMO. One of the reasons for the lack of correlation in the detrended data could be due to the small ratio between the consecutive yearly variabilities and the long-term linear trends (about 0.1) for GT, PDO or AMO (Figure S4).

How can the global warming trend explain 67% of the variance in the global cloud cover trends and the AMO trend explain 49% (line 158)? You can't explain more than 100% of the variance, unless the indices are not independent of one another. In other words, it doesn't appear that the global warming, PDO, and AMO indices are actually orthogonal to one another (as is claimed on lines 166-167).

We agree there is a problem of explaining more than 100% of the variance. We didn't try to hide the problem, as we stated in the original manuscript: "PDO together with AMO and GT, obviously has a problem of over 100% explanation of the spatial variabilities of linear trends in cloud cover and precipitation. Since the trend of global SST has been removed from the PDO and AMO indexes in this study, in theory GT should be orthogonal to those of PDO and AMO." In practice the orthogonality is not attained because the trend of global SST doesn't equal to the real influence of global temperature on PDO or AMO. It is difficult to remove exactly the influence of global temperature from PDO or AMO index. This is likely the main reason of the problem of over 100% explanation.

4) The authors are examining cloud and precipitation features in the deep tropics and attributing them to a poleward shift in the Hadley cell edge and midlatitude jet streams (lines 131-132, 138-140). The expansion of the Hadley cell and poleward shift of the jet streams affects precipitation in the subtropics and midlatitudes (poleward of 30 degrees latitude), not in the deep tropics. For tropical precipitation changes, the authors need to really be comparing their results with recent changes in the ascending branch of the Hadley cell (Intertropical Convergence Zone), not the descending branch in the subtropics.

Thanks for an excellent point! In the Figure 2e (in our response to referee 3), one can see that the expansion of the Hadley cell as measured by clouds starts at belt 2 (3.75º latitude), i.e. the blue curve starts to move to the right of the black curve near 3.75º latitude. This is near the center of the ascending branch of the Hadley cell in the Maritime Continent. The expansion of the Hadley cell as measured by precipitation

(Figure 2a) starts near belt 5 (12.5º latitude). This is likely due to the constraint on the change of global total annual precipitation, which is equal to global evaporation and determined by the global surface energy budget, increases with global temperature at a rather small rate of about 2%–3% $K^{-1}$ (Cubasch et al., 2001).

5) Section 3b seems like a separate study and to not be related to the rest of the paper. Trends in a small region are not necessarily affected by global drivers, and regional influences are not discussed at all. This data analysis also suffers from similar problems as the global analyses in section 3a (see major comments #2 and #3).

All three referees raised this important concern. We have made changes in both the abstract and the beginning of section 3.2 to better connect the global part and the analysis of data in China (see below). Moreover, we now have established a more consistent results for the two parts.

The new addition to section 3.2 is: The global analysis is extended by investigating connections between clouds and precipitation in China, which has a large number of long-running, high-quality surface weather stations over the period of 1957–2005. The long-running data enable the analysis to be carried out over a period that AMO loses while PDO flips its linear trend. More importantly, the high-quality data allow us to make a meaningful analysis without using the correlation method, which has an intrinsic weakness in implying a cause-effect relationship as discussed above.

The revision to the abstract on this issue is: The global analysis is extended by investigating connections between clouds and precipitation in China, which has a large number of long-running, high-quality surface weather stations in 1957–2005, which reveals a quantitative matching relationship between the reduction in light precipitation and the reduction of total cloud cover. Furthermore, our study suggests that the reduction of cloud cover in China is primarily driven by the global temperature conditions, PDO plays a secondary role, while the contribution from AMO and Niño3.4 is insignificant, consistent with the global analysis.

Minor Revisions Lines 20-29: The trends described in this paragraph do not appear to closely match those shown in Norris et al. (2016), especially over land and over the Indian Ocean.

We are confused by this comment. We checked and compared Figure 1 in Norris et al. (2016) with our Figure 1, they are very consistent.

Lines 54-71: Somewhere in this paragraph, it is probably worth mentioning that the constraint on global precipitation is 2–3% per K, and not 7% per K. See, for example, Jeevanjee and Romps (2018; https://doi.org/10.1073/pnas.1720683115).

Agree, this is now added in two places. One is in the 3rd paragraph of section 3.1, the other in the 7th paragraph of the same section.

Line 69, 131-132, 138-140: See major comment #4. The expansion of the Hadley cells has nothing to do with enhancement of tropical precipitation. It is related to subtropical static stability (Chemke and Polvani 2019: https://doi.org/10.1175/JCLI-D-18-0330.1). If anything, an expansion of tropical precipitation would contradict the literature, which suggests a narrowing of the Intertropical Convergence Zone in a warming climate (Byrne and Schneider 2016: https://doi.org/10.1002/2016GL070396; Su et al. 2017: https://www.nature.com/articles/ncomms15771).

We understand this is a controversial point. Please see our response to your major comment #4.

Line 160: The figure for the PDO really belongs in the main body of the paper, as it is part of the main conclusions of the paper (see abstract).

Thanks, we now have two figures (Figures 2d and 2h in our response to referee 3, i.e. Figures 3d and 3h in our revised manuscript) for the PDO.

Line 187: No, the key difference here is that Chen et al. (2019) use the first 300 years of control model simulations to define the cloud cover patterns associated with the PDO

and AMO, which avoids the issues of concurrent trends in the indices using the observations (see major comment #3 above).

We disagree on this point, because we question the credibility of climate models in the simulation of changes in clouds and precipitation as a function of AMO or PDO.

Lines 189-193: Why is the PDO deemed insignificant here? Is this based entirely on Eastman and Warren's analysis? Nothing shown in this paper appears to make the PDO less significant than the AMO (see Table 1).

Please see our response to your major comment#3. The new results on the widening of the Hadley circulation (Figures 2a-2h in our response to referee 3) suggest that the contribution of both PDO and AMO are insignificant compared to the global temperature increase.

Lines 208-210: Could the increase in non-precipitation days and decrease in light precipitation days reflect a change in reporting method? How do you know that these changes are in fact physical?

Trenberth et al. (2003) summarized the global warming hypothesis by explaining that the precipitation intensity of storms should increase at about the same rate as atmospheric moisture, which is about 7% $K^{-1}$ according to the Clausius–Clapeyron equation. The precipitation intensity could even exceed the 7% $K^{-1}$ because additional latent heat released from the increased water vapour could invigorate the storm and pull in more moisture from the boundary layer, forming a positive feedback cycle (i.e. the moisture-convection-latent heat feedback cycle) and leaving less moisture available for light and moderate precipitation.

Lines 237: Difficult to read as written. The equation should be spaced out. Figures: I would suggest inverting the color bar such that blues correspond to more clouds/precipitation and reds correspond to less.

Thanks for the suggestion. After some deliberation we choose to retain the current color

bar.

Table 1: How are you evaluating significance? I have a difficult time believing that a correlation of 0.02 is still significant at the 95% confidence level. Are you taking into account autocorrelations among neighboring grid points, which would greatly reduce the number of degrees of freedom in your t-test? Table 2: Similarly, how is significance being evaluated here? A trend of 0% (see T60%) should not be statistically significant at all, especially at the 99% level.

We used the function imbedded in R named corr to do this significance test. The function corr we chose applies Pearson correlation formula:

$$r = \frac{\sum(x - m_x)(y - m_y)}{\sqrt{\sum(x - m_x)^2 \sum(y - m_y)^2}}$$

$m_x$ and $m_y$ are the means of x and y variables.

The p-value of the correlation is determined by calculating the t value as follow:

$$t = \frac{r}{\sqrt{1 - r^2}}\sqrt{n - 2}$$

then using t distribution table for the degrees of freedom: df = n-2 to get the p-value.

We believe even when the correlation coefficient r is very small, due to the big value of n (the number of samples we used in calculation), the t value should remain a very big value, therefore brings a reliable significance.

Typos Line 20: are of great importance

Thanks, changed accordingly.

Line 27: places affiliated to Australia – not sure what this means, please rephrase

Rephrased to "around Australia".

Line 98: provided by

Changed accordingly.

Line 99: retained

Changed accordingly.

Line 105-106: Incomplete sentence . . . please rewrite.

Rewritten accordingly.

Line 145: is robust

This part is rewritten.

Figure 6a: bottom 10%-40%

Changed accordingly.

---

## Author Response (AR1)

**Dear Editor,**

We appreciate the prompt review and would like to thank the three Reviewers' perceptive and helpful comments and suggestions on our manuscript entitled "Observed Trends of Clouds and Precipitation (1983–2009): Implications for Their Cause(s)", Author(s): Xiang Zhong et al., MS No.: acp-2020-577, MS type: Research article. We have carefully considered all comments and suggestions and carried out major revisions as suggested. The sequence of our responses is ordered by the time received, i.e. Referee #3 followed by Referee #1 and Referee #2. We believe that the revisions have resulted in a significantly improvement of the paper. Listed below are point-by-point responses to all comments and suggestions of the three reviewers (Reviewer's points in black, our responses in blue).

**Anonymous Referee #3 interactive comment**

The authors present two analyses concerning trends in clouds and rainfall. One uses global, satellite-observed cloud and precipitation data to show that cloud cover and precipitation trends are consistent with an expanding tropical belt. The other looks at surface-observed clouds and rain rates in China to show that light, stratiform rain and overcast clouds are declining while convective rain associated with more broken clouds is relatively more common. These results are consistent with prior work showing a widening tropical belt and a trade-off from stratiform precipitation in favor of convective precipitation.

The work addresses some very large and interesting problems using a fairly simple and easy to understand method, which is commendable. The quality and presentation of the manuscript is high and the work presents great value to the community. There are a few places where the analysis needs a bit more rigor, especially regarding the removal of long-term variation from timeseries in the correlation analysis. It is crucial that we know that the correlations we see are due to interannual variations and not due to coinciding trends. If the authors can do this bit of extra work, the results will be significantly more robust. We appreciate very much for these encouraging comments. As shown below, we have made extensive revisions in point-by-point responses to your comments and suggestions.

**Major comments:**

There is talk of a widening Hadley cell, and the results do hint at this, but I would love to see a bit more rigor in 1) defining what your data show as the tropical belt, maybe with a zonal mean plot showing the mean clouds/precipitation for latitude zones, then 2) showing the mean trends for the same zones. You could do this globally, or for a specific region between longitude bounds.

We gratefully accept this suggestion by explicitly evaluating the widening of Hadley cell in the observed trends of precipitation and cloud cover "for a specific region between longitude bounds". The results reveal a pleasant surprise, as Figure 3 below provides adequate evidence to show that the trend of global temperature, rather than the trends of AMO and PDO, is the primary contributor to the observed linear trend of precipitation in 1983–2009.

As a measure of the widening of Hadley circulation, we calculate and illustrate the expansion of cloud cover and precipitation as a function of 16 rectangle belts centered in the middle of Kalimantan, Indonesia which is located near the major ascending/wet zone of Hadley cell (Figure 2). Each rectangle belt is 2.5 degree wide in both latitude and longitude except the first rectangle is 5 degree wide in latitude and 55 degree wide in longitude.

Figure 2. Maps of the 16 rectangle belts of 2.5 degree wide in both latitude and longitude centered in the middle of Kalimantan, Indonesia which is located near the major ascending/wet zone of Hadley cell. The expansion of cloud cover and precipitation relative to these belts are used as a measure of the widening of Hadley circulation.

Figures 3a and 3e depict for annual precipitation and total cloud cover, respectively, their "climatology" (black curve) and "climatology + change during 1983-2009" (blue curve). It can be seen that, for a specific value of the y-axis, the blue curve is characterized by a shift horizontally (x-axis direction) to the right (i.e. higher number of belt) of the black curve for most of Figures 3a and 3e. In comparison, there is very little upward shift in the vertical or y-axis direction, especially at low-end (belt 1 and 2) and high-end belts (belt 15 and 16). As a result, there is hardly any enhancement in total cloud cover and total precipitation. These characteristics can be interpreted as an expansion to higher latitudes and wider longitudes, i.e. widening of the Hadley and Walker circulations during the period of 1983-2009. Quantitatively the degree of

expansion depends on the selected value of the y-axis, increasing quickly when the value is near 1000mm precipitation level (Figure 3a) or 55% of TCC (Figure 3e). The value of shift is typically within the range of one quarter to three quarters of a belt width (2.5 degree), or about 0.6-1.9 degree. These annual values are comparable to the poleward shift of the subtropical dry zones (up to 2° decade-1 in June - July - August (JJA) in the Northern Hemisphere and 0.3–0.7° decade-1 in June - July - August and September - October - November in the Southern Hemisphere) found by Zhou et al. (2011).

---

## Author Response (AR2)

Dear Editor,

We appreciate the two Reviewers' perceptive and helpful comments and suggestions on our manuscript entitled "Observed Trends of Clouds and Precipitation (1983–2009): Implications for Their Cause(s)" (MS No.: acp-2020-577). We have carefully considered all comments and suggestions and carried out major revisions as suggested. We believe that the revisions have resulted in a significantly improvement of the paper. Listed below are point-by-point responses to all comments and suggestions of the two reviewers (ordered by the time of review came to us). Reviewer's points are in black, our responses in blue.

**Anonymous Referee #2**

I have read through the revised paper, and find that many of the major issues originally raised by the reviewers still largely remain in the manuscript.

1. Reviewer 3 stated that "the authors need to show that the relationships between global temperature and regional variations in cloud cover and precipitation are consistent when linear trends are removed." The authors have done this analysis in Table S1, and found that this is actually not the case. In fact, they conclude that the "high correlation coefficients are nearly entirely contributed by the long-term linear trends." As I stated in my original review, just because global temperatures are warming, it doesn't mean that concurrent trends in clouds and precipitation are necessarily caused by global warming. The similarity in Figs. 1 and 4 is by construction, as the global temperature time series is dominated by an increasing trend (so any trend in clouds and precipitation will by definition be highly correlated with global temperature). While the authors have acknowledged this problem at various points in the manuscript (for example, see paragraph starting on line 325), the large amounts of variance attributed to global warming, AMO, and PDO are still discussed throughout as a key conclusion of the manuscript (line 11). A much more careful analysis (such as that discussed in Chen et al. 2019) needs to be performed to more precisely partition the recent trends into the

global warming, PDO, and AMO components.

**Response:**

We worked very hard on the first revision, here we hope the second revision would get this reviewer's approval. We fully agree that a good correlation of concurrent trends in clouds and precipitation with the trends of global warming, PDO, or AMO does not imply any cause-effect relationship, thus cannot be used to partition the recent trends in clouds and precipitation into the global warming, PDO, and AMO components. As this reviewer also noted that this point was stated repeatedly in our original as well as the first revised manuscripts. It was a major reason we decided to focus our analysis on two critical regional characteristics of the trends in clouds and precipitation: namely the widening of the global Hadley and Walker circulations (see below changed to: the broadening of the major ascending zone of Hadley circulation) and the long-running trends in the high quality station data of clouds and precipitation in China, to help partitioning the recent trends in clouds and precipitation into the global warming, PDO, and AMO components. Surely as this reviewer stated that regional results might not be representative of the global phenomena, nevertheless the partitioning established globally should be evaluated more rigorously regionally. In addition, the area of the rectangles in Fig. 2 covers about one third of the entire domain in this study and it includes most of the prominent features in the trends in clouds and precipitation. In regard to the analysis of clouds and precipitation in China, we have the advantage of the large number of long-running, high-quality surface weather stations over the period of 1957–2005 (1957–2017 for precipitation). The long-running data enable the analysis to be carried out over a period in which the linear trends of AMO and PDO have both diminished to insignificant values. More importantly, the high-quality data allow us to make some critical analyses without using the correlation method, which has an intrinsic weakness in implying a cause-effect relationship as discussed above.

We have used a straightforward arithmetic analysis of the relationship between interannual variabilities in cloud cover and light precipitation in China, which provides

evidence of a quantitatively matching closure between the variabilities of light precipitation and those of cloud cover. Furthermore, the cause-effect relationship between the changes in precipitation intensity and global warming has been investigated in a large number of studies which include theoretical, modeling as well as correlational approaches (e.g. Trenberth, 1998; Allen and Ingram, 2002; Trenberth et al., 2003; Sun et al., 2007; Liu et al., 2015, 2016). These studies concluded that the extensive worldwide reports of enhancements in heavy precipitation and reductions in the light and moderate precipitation are most likely a result of global warming and the primary driving mechanism is the moisture-convection-latent heat feedback cycle associated with global warming.

We choose not to follow the modeling analysis used by Chen et al (2019) because current climate models tend to have large uncertainties, particularly in the simulation of regional distributions of clouds and precipitation, as evident by the low model performance rating during the IPCC model evaluation (Flato et al., 2013).

2. All three reviewers found the suggested linkage of the results with Hadley cell expansion to be inadequately supported and recommended further analyses. To address these comments, the authors have added two figures (Figs. 2-3) to describe a broadening of high cloud cover and precipitation over the deep convective region over the western Pacific Ocean centered on Indonesia. The authors conclusively show that this region has expanded in recent decades, which is an interesting new result. However, the authors attribute this very local feature to a widening of the global Hadley and Walker circulations (line 146), but do not show any evidence to support this claim. For example:

- Figures 2 and 3 combine changes in the zonal and meridional directions, making it impossible to tell which direction is most contributing to the changes.

 - No analysis is provided for the eastern tropical Pacific Ocean. To show that the Walker circulation is widening, the authors need to show that the descending branch is not changing. Figure 3 only shows changes in the ascending branch.

- The Hadley cell is a zonal-mean quantity. Changes in the ascending region over the western Pacific Ocean do not necessarily imply changes in the zonal mean.

- To show that the Hadley cell is widening, the authors need to show that the descending branch is moving. Even if the deep tropical precipitation is broadening, it doesn't necessarily imply that the descending branch is moving.

Because of these two major issues remaining in the manuscript, I cannot recommend publication at this time.

**Response:**

Both reviewers raised this concern on the widening of Hadley circulation and recommended changing some of the basic terminology used. We acknowledge that what we showed in Fig. 3 of the first revision should be more precisely described as the broadening/expansion of the major ascending/wet zone of Hadley circulation, rather than widening of the Hadley circulation. Therefore, we decide to accept both reviewers' suggestion and change "the widening of Hadley circulation" to "the broadening of the major ascending zone of Hadley circulation" throughout the paper. Having said that, we would like to explain that, in our view, Hadley and Walker cells are two components of one single atmospheric circulation; and the expanding/broadening convective region over the western Pacific Ocean within the rectangles in Figure 2 is by far the predominant ascending branch of Hadley and Walker cells, which encompasses as much as one third of the entire domain in this study. As stated in our first revision (Lines 155–157), Zhou et al. (2011) had already shown that the broadening of this ascending branch of Hadley and Walker cells is a primary contributor to the widening of Hadley cell. That is why we misnamed "the broadening of the major ascending zone of Hadley circulation" as "the widening of Hadley circulation".

Line 11: "and negligible" --- awkward phrasing … I would remove referring to ENSO in the abstract if it's negligible.

**Response:**

Done as suggested.

Line 166: rectangle

**Response:**

Corrected.

Lines 167, 210: I don't see the close correspondence between Figs. 3e and 3f.

**Response:**

We have clarified this point by revising the text near Lines 208–215 to "The quantity of global total annual precipitation, which is equal to global evaporation and determined by the global surface energy budget, increases with global temperature at a rather small rate of about 2%–3% $K^{-1}$ (Cubasch et al., 2001), which is manifested in Figs. 3a and 3b by the small/negligible change of the net area between blue and black lines, while Figs. 3c and 3d have significant enhancements. Therefore, based on the results of Figs. 3a-3d, we propose that the trend in global temperature, rather than that of AMO and PDO, is the primary contributor to the observed linear trend of precipitation in 1983–2009. Likewise, Figs. 3e and 3f both have small/negligible changes of the net areas between blue and black lines, while Figs 3g and 3h have significant enhancements of cloud cover. Accordingly, we propose that the trend in global temperature, rather than that of AMO and PDO, is the primary contributor to the observed linear trend of cloud cover in 1983–2009."

Lines 213-214: Incorrect figure (Fig. 2e) is cited here.

**Response:**

Sorry, changed to Fig. 3e.

Line 242: AMO loses? … not sure what is meant by this, please rephrase

**Response:**

Thanks. The associated sentence near Line 249 has been rephrased to "The long-running data enable the analysis to be carried out over a period in which the linear trends of AMO and PDO have both diminished to insignificant values."

Line 247: Figure 6

**Response:**

Done.

Line 259: Fig. 7a

**Response:**

Done.

Line 266: Fig. 7b

**Response:**

Done.

Line 270: Fig. 8

**Response:**

Done.

Figure 2 caption: Need to clarify that these maps are showing cloud cover trends from Fig. 1.

**Response:**

Done as suggested.

Statistical significance in Tables 1 and 2: As per my previous comment, the authors

shouldn't be using a sample size of n to calculate the p-value of the correlation coefficient, but rather an effective sample size (n*) that accounts for the autocorrelation in the time series. See equation 31 of Bretherton et al. (1999): https://doi.org/10.1175/1520-0442(1999)012<1990:TENOSD>2.0.CO;2

**Response:**

Sorry for this mistake. Per your suggestion, we followed Bretherton et al. (1999) to recalculate the effective sample size ($N_{ef}^*$) and perform the statistical significance test. Now the Table 1 is revised as follows:

| R | Trend of TCC | Trend of TP |
|---|---|---|
| $\delta$(GT) | 0.82 *** | 0.93 *** |
| $\delta$(-PDO) | 0.62 *** | 0.73 *** |
| $\delta$(AMO) | 0.70 *** | 0.77 *** |
| $\delta$(Niño3.4) | -0.20 *** | 0.02 |
| $\delta$(GT)+$\delta$(-PDO) | 0.74 *** | 0.85 *** |
| $\delta$(GT)+$\delta$(AMO) | 0.86 *** | 0.89 *** |
| $\delta$(GT)+$\delta$(Niño3.4) | 0.89 *** | 0.93 *** |
| $\delta$(-PDO)+$\delta$(AMO) | 0.67 *** | 0.79 *** |
| $\delta$(-PDO)+$\delta$(Niño3.4) | 0.61 *** | 0.72 *** |
| $\delta$(AMO)+$\delta$(Niño3.4) | 0.65 *** | 0.73 *** |
| $\delta$(GT)+$\delta$(-PDO)+$\delta$(AMO) | 0.76 *** | 0.87 *** |
| $\delta$(GT)+$\delta$(-PDO)+$\delta$(Niño3.4) | 0.72 *** | 0.84 *** |
| $\delta$(GT)+$\delta$(AMO)+$\delta$(Niño3.4) | 0.86 *** | 0.88 *** |
| $\delta$(-PDO)+$\delta$(AMO)+$\delta$(Niño3.4) | 0.65 *** | 0.78 *** |
| $\delta$(GT)+$\delta$(-PDO)+$\delta$(AMO)+$\delta$(Niño3.4) | 0.75 *** | 0.86 *** |

Note: GT denotes global temperature anomalies. $\delta$(GT) denotes $\Delta GT \times dTCC/d(GT/GT\sigma)$ or $\Delta GT \times dTP/d(GT/GT\sigma)$, where $\Delta GT$ is the change of GT for the studied period and $GT\sigma$ is the standard deviation of GT, and other factors likewise. *** indicates statistically significant at the 99% confidence level based on student's t test.

For the comment "A trend of 0% (see T60%) should not be statistically significant at all", sorry for our carelessness, we checked our calculation again and found that we mislabeled 0 with ***. The corresponding p-value was 0.75. Table 2 is revised as follows:

Table 2. Climatology and days changed for precipitation days and cloudy days

| | Climatology (day) | Change rate (day per decade) | Relative change rate (% per decade) | Change over 49 years (day) | Relative change over 49 years (%) |
|---|---|---|---|---|---|
| NPD | 202.5 | 4.5±0.2 *** | 2.2±0.1 *** | 22.1±1.0 *** | 10.9±0.5 *** |
| B10% | 116.9 | -4.2±0.2 *** | -3.6±0.2 *** | -20.6±1.0 *** | -17.6±1.0 *** |
| B20% | 132.0 | -4.3±0.2 *** | -3.3±0.2 *** | -21.1±1.0 *** | -16.0±1.0 *** |
| B30% | 141.2 | -4.4±0.2 *** | -3.1±0.1 *** | -21.6±1.0 *** | -15.3±0.5 *** |
| B40% | 147.5 | -4.5±0.2 *** | -3.1±0.1 *** | -22.1±1.0 *** | -15.0±0.5 *** |
| T60% | 15.0 | 0±0 | 0±0 | 0±0 | 0±0 |
| CFD | 34.9 | 2.3±0.1 *** | 6.6±0.3 *** | 11.3±0.5 *** | 32.3±1.5 *** |
| ≤50% | 152.3 | 4.3±0.2 *** | 2.8±0.2 *** | 21.1±1.0 *** | 13.7±1.0 *** |
| >50% | 212.7 | -4.3±0.2 *** | -2.0±0.2 *** | -21.1±1.0 *** | -9.9±1.0 *** |

Note: *** indicates statistically significant at the 99% confidence level based on student's *t* test. NPD denotes non-precipitation days, B10% denotes bottom 10% precipitation days, T60% denotes top 60% precipitation days, ≤50% denotes ≤50% cloud cover days and CFD denotes cloud-free days.

We have added corresponding description on Lines 116–117.

Table S2 is not discussed at all in the main text.

**Response:**

Table S2 is deleted.

**Anonymous Referee #1**

I commend the authors for the significant work in revising their manuscript. Overall, it has been substantially improved. However, there still seems to be a bit of disconnect in some of their statements, particularly those related to what is meant by expansion of the Hadley cell.

To summarize, I feel as if the authors are not using correct terminology, as has been previously established in (many) prior publications. Which adds confusion and makes interpretation of the results more difficult than is necessary. I would recommend changing some of the basic terminology used throughout the paper, such that it is consistent with prior publications.

Comments

As mentioned in the first round of reviews, Hadley cell expansion refers to a poleward displacement of the outer edge of the circulation. There are numerous publications that have used this definition. In the context of precipitation, it has been defined as a poleward shift of the subtropical latitude where P-E = 0; the subtropical latitude where P is a minimum could also work. It has also been pointed out that looking at this separately in each hemisphere is important, because the NH and SH show different tropical expansion signals.

The authors choose to construct their own methodology to define "tropical expansion", based on the rectangular boxes. That's fine. But they do not really give a simple explanation for what this new methodology is calculating. It appears to me, based on panel a in Figure 3 from the response, that precipitation is not changing near the center of the region (inner rectangles), but it is increasing in the outer regions (outer rectangles). I assume precipitation is estimated in each rectangle independently, and not summed over the inner rectangles? In any case, this would seem to suggest an increase in tropical precipitation starting near layer 6 and extending outwards to layer 15. In other words, tropical precipitation in this region is moving outwards from the center

rectangle? Or maybe a better description is that there is just an increase in tropical precipitation moving outwards from the local maximum (assuming the inner rectangle is a local maximum)? This is not "tropical expansion" as used in numerous other publications. To add confusion, the authors also describe this as not only Hadley cell expansion, but also expansion of the Walker circulation.

The center of the rectangles is "in the middle of Kalimantan, Indonesia which is located near the major ascending/wet zone of Hadley cell". What is meant by "near" the major wet zone? Is the center rectangle chosen so that it represents a local maximum in precipitation?

The authors then go on to say (again, in the response): "In summary, the spatial distributions of the linear trends of total cloud cover and precipitation are characterized primarily by a widening of the center of precipitation (ascending/wet zone of Hadley cells) over the Maritime Continent in all directions".

Yes, I agree with this description. But this is not what is meant by "tropical expansion". This is more related to the thickness of the band of intense tropical rain, right? And the authors are showing an increase in this "thickness"? Perhaps this is not related to "tropical expansion". But it does seem to be interesting, as others have shown that under continued GHG increases, the ITCZ is projected to narrow (or decrease its "thickness"). But the authors are showing the opposite. It would appear that the authors should remove "tropical expansion" type statements and verbiage, and instead replace this with something more similar to what they are quantifying—"thickness" or area, etc. of the intense precipitation over the Maritime continent.

I think the last reviewer summarized this concern well:

Line 69, 131-132, 138-140: See major comment #4. The expansion of the Hadley cells has nothing to do with enhancement of tropical precipitation. It is related to subtropical static stability (Chemke and Polvani 2019: https://doi.org/10.1175/JCLI-D-18-0330.1). If anything, an expansion of tropical precipitation would contradict the literature, which

suggests a narrowing of the Intertropical Convergence Zone in a warming climate (Byrne and Schneider 2016: https://doi.org/10.1002/2016GL070396; Su et al. 2017: https://www.nature.com/articles/ncomms15771).

Sorry for the confusion, we should have said that our result on the broadening of the major ascending zone of Hadley circulation applies only to the rectangles circling the major ascending zone of Hadley circulation. We also noticed significant contraction of the ITCZ between 80ºW–180ºW, as shown in Fig. 1b.

But as can be seen, the authors have not really addressed this confusion. They continue to refer to their signal as "tropical expansion" (as well as expansion of the Walker circulation). Furthermore, their response again seems to be disconnected from the comment. The comment is pointing out that the authors are not using the term "tropical expansion" properly. And instead of changing it, they simply say that this is controversial? I don't quite follow.

**Response:**

Both reviewers raised this concern on the widening of Hadley circulation and recommended changing some of the basic terminology used. We acknowledge that what we showed in Fig. 3 of the first revision should be more precisely described as the broadening/expansion of the major ascending/wet zone of Hadley circulation, rather than widening of the Hadley circulation. Therefore, we decide to accept both reviewers' suggestion and change "the widening of Hadley circulation" to "the broadening of the major ascending zone of Hadley circulation" throughout the paper. Having said that, we would like to explain that, in our view, Hadley and Walker cells are two components of one single atmospheric circulation; and the expanding/broadening convective region over the western Pacific Ocean within the rectangles in Figure 2 is by far the predominant ascending branch of Hadley and Walker cells, which encompasses as much as one third of the entire domain in this study. As stated in our first revision (Lines 155–157), Zhou et al. (2011) had already shown that the broadening of this ascending

branch of Hadley and Walker cells is a primary contributor to the widening of Hadley cell. That is why we misnamed "the broadening of the major ascending zone of Hadley circulation" as "the widening of Hadley circulation".

About the calculation for Fig. 3, the reviewer's understanding is correct. To make this easily understandable, we have added "The summing up was done for each rectangle independently, the inner rectangles were not included." on Lines 141–142. The center rectangle (5 degree wide in latitude and 55 degree wide in longitude) was chosen because it locates near the major ascending zone of Hadley cell which also coincides with the local wet zone.

Regarding this response:

Direct effects of anthropogenic aerosols on clouds and precipitation tend to be regional and/or sub- yearly time scale, which are beyond the scope of discussion in this study

Obviously, this is a small point and not particularly important to the main analysis. But again, there seems to be a disconnect in the response. Why are possible aerosol effects on cloud/precipitation important on only sub-yearly time scales? I do not think this is true. There have been multi-decadal changes in anthropogenic aerosol emissions, which leads to multi-decadal changes in aerosol forcing. So it would stand to reason that such a multi-decadal forcing may in fact lead to long term changes in temperature, clouds, precipitation, etc.

It's a bit odd (and frustrating) that I've pointed out past papers that have addressed causes of tropical expansion (e.g., aerosols, as well as the PDO), and the authors have chosen to disregard any acknowledgement of these prior papers. Why not add a simple sentence in the introduction that points out prior papers that have addressed the causes of tropical expansion? But then again, I do not think what the authors show is really "tropical expansion", and therefore, the causes of their signal may very well have nothing to do with previously identified causes of tropical expansion. So in a round-about way, lack of addressing this point fine, I suppose.

**Response:**

In the original manuscript as well as the first revision, we did address the issue of "There have been multi-decadal changes in anthropogenic aerosol emissions, which leads to multi-decadal changes in aerosol forcing". This can be seen in Lines 235–236 "The long-term radiative effect of aerosols on the global temperature and other climate parameters are expected to be imbedded in the observed changes of these climate parameters, and thus included in this study."

The effects of anthropogenic aerosols on clouds and precipitation by acting as cloud condensation nuclei (CCN) is a highly controversial issue. We apologize for our timidity in trying to deal with this issue in earlier manuscripts. In this revision, this issue is addressed by adding Lines 234–239: "The effects of anthropogenic aerosols on clouds and precipitation by acting as cloud condensation nuclei (CCN) is a highly controversial issue which has been discussed extensively in a number of studies as well as one of our earlier papers (Liu et al., 2015). We defer the discussion on this issue to future studies, and acknowledge here that the CCN effects could introduce an unknown amount of uncertainty in this study." In addition, we elaborate below the controversy of this issue by quoting a key paragraph from Liu et al. (2015):

"It has been long recognized that aerosols may have a significant influence on clouds and precipitation by acting as cloud condensation nuclei (Warner and Twomey, 1967; Albrecht, 1989; Ramanathan et al., 2001; Andreae et al., 2004; Dai et al., 2008; Koren et al., 2008). The aerosol effect on precipitation processes, considered part of the "Albrecht" effect—the "second indirect" effect on cloud extent and life time (Ackerman et al., 1978; Albrecht, 1989; Hansen et al., 1997)—is complex and uncertain, especially for mixed-phase convective clouds (Tao et al., 2012). There have been numerous studies conducted on the effects of aerosol on total precipitation over different periods (e.g. annual, seasonal), producing mixed results. An excellent example is Warner (1971), who concluded there was no change in 60 years of precipitation due to aerosols emitted from sugarcane burning in northern Australia. In addition, a report

by the U.S. National Research Council (2003) concluded "there still is no convincing scientific proof of the efficacy of international weather modification efforts," of which many are modification efforts using aerosols."

The authors again use this terminology:

Further analysis of the widening of the Hadley and Walker circulations (Figures 3a-3h)

What is meant by widening of the Walker circulation?

The authors also use this description of their signal, which is more reasonable in my opinion: "widening of the center of precipitation over the Maritime Continent in all directions". And maybe "broadening" in more appropriate than "widening", as widening implies an increase in one spatial direction, whereas broadening is more general.

**Response:**

This point has been addressed in our response to this reviewer's first comment.

Another example of a disconnect between reviewer comment and author response pertains to significance:

Table 1: How are you evaluating significance? I have a difficult time believing that a correlation of 0.02 is still significant at the 95% confidence level. Are you taking into account autocorrelations among neighboring grid points, which would greatly reduce the number of degrees of freedom in your t-test? Table 2: Similarly, how is significance being evaluated here? A trend of 0% (see T60%) should not be statistically significant at all, especially at the 99% level.

We used the function imbedded in R named corr to do this significance test. The function corr we chose applies Pearson correlation formula…The p-value of the correlation is determined by calculating the t value as follow…then using t distribution table for the degrees of freedom: $df = n-2$ to get the p-value.   We believe even when

the correlation coefficient r is very small, due to the big value of n (the number of samples we used in calculation), the t value should remain a very big value, therefore brings a reliable significance.

Yes, the small correlations (e.g., 0.02) are deemed significant in this analysis likely due to the large n. But this comment is suggesting n is not as large as what the authors are using, due to spatial autocorrelation. But the authors do not address this point.

**Response:**

Sorry for this mistake. As we have addressed in the response to reviewer #2's comment, we followed Bretherton et al. (1999) to recalculate the effective sample size ($N_{ef}^*$) and perform the statistical significance test. Now the Table 1 is revised as follows:

| R | Trend of TCC | Trend of TP |
|---|---|---|
| $\delta$(GT) | 0.82 *** | 0.93 *** |
| $\delta$(-PDO) | 0.62 *** | 0.73 *** |
| $\delta$(AMO) | 0.70 *** | 0.77 *** |
| $\delta$(Niño3.4) | -0.20 *** | 0.02 |
| $\delta$(GT)+$\delta$(-PDO) | 0.74 *** | 0.85 *** |
| $\delta$(GT)+$\delta$(AMO) | 0.86 *** | 0.89 *** |
| $\delta$(GT)+$\delta$(Niño3.4) | 0.89 *** | 0.93 *** |
| $\delta$(-PDO)+$\delta$(AMO) | 0.67 *** | 0.79 *** |
| $\delta$(-PDO)+$\delta$(Niño3.4) | 0.61 *** | 0.72 *** |
| $\delta$(AMO)+$\delta$(Niño3.4) | 0.65 *** | 0.73 *** |
| $\delta$(GT)+$\delta$(-PDO)+$\delta$(AMO) | 0.76 *** | 0.87 *** |
| $\delta$(GT)+$\delta$(-PDO)+$\delta$(Niño3.4) | 0.72 *** | 0.84 *** |
| $\delta$(GT)+$\delta$(AMO)+$\delta$(Niño3.4) | 0.86 *** | 0.88 *** |
| $\delta$(-PDO)+$\delta$(AMO)+$\delta$(Niño3.4) | 0.65 *** | 0.78 *** |
| $\delta$(GT)+$\delta$(-PDO)+$\delta$(AMO)+$\delta$(Niño3.4) | 0.75 *** | 0.86 *** |

Note: GT denotes global temperature anomalies. $\delta$(GT) denotes $\Delta GT \times dTCC/d(GT/GT\sigma)$ or $\Delta GT \times dTP/d(GT/GT\sigma)$, where $\Delta GT$ is the change of GT for the studied period and $GT\sigma$ is the standard deviation of GT, and other factors likewise. *** indicates statistically significant at the 99% confidence level based on student's t test.

For the comment "A trend of 0% (see T60%) should not be statistically significant at

all", sorry for the carelessness, we checked our calculation again and found that we mislabeled 0 with ***. The corresponding p-value was 0.75. Table 2 is revised as follows:

Table 2. Climatology and days changed for precipitation days and cloudy days

|  | Climatology (day) | Change rate (day per decade) | Relative change rate (% per decade) | Change over 49 years (day) | Relative change over 49 years (%) |
|---|---|---|---|---|---|
| NPD | 202.5 | 4.5±0.2 *** | 2.2±0.1 *** | 22.1±1.0 *** | 10.9±0.5 *** |
| B10% | 116.9 | -4.2±0.2 *** | -3.6±0.2 *** | -20.6±1.0 *** | -17.6±1.0 *** |
| B20% | 132.0 | -4.3±0.2 *** | -3.3±0.2 *** | -21.1±1.0 *** | -16.0±1.0 *** |
| B30% | 141.2 | -4.4±0.2 *** | -3.1±0.1 *** | -21.6±1.0 *** | -15.3±0.5 *** |
| B40% | 147.5 | -4.5±0.2 *** | -3.1±0.1 *** | -22.1±1.0 *** | -15.0±0.5 *** |
| T60% | 15.0 | 0±0 | 0±0 | 0±0 | 0±0 |
| CFD | 34.9 | 2.3±0.1 *** | 6.6±0.3 *** | 11.3±0.5 *** | 32.3±1.5 *** |
| ≤50% | 152.3 | 4.3±0.2 *** | 2.8±0.2 *** | 21.1±1.0 *** | 13.7±1.0 *** |
| >50% | 212.7 | -4.3±0.2 *** | -2.0±0.2 *** | -21.1±1.0 *** | -9.9±1.0 *** |

Note: *** indicates statistically significant at the 99% confidence level based on student's $t$ test. NPD denotes non-precipitation days, B10% denotes bottom 10% precipitation days, T60% denotes top 60% precipitation days, ≤50% denotes ≤50% cloud cover days and CFD denotes cloud-free days.

**References:**

Ackerman, A. S., and Coauthors: Summary of METROMEX, Volume 2: Causes of Precipitation Anomalies. Illinois State Water Survey, Urbana, Bulletin 63, 399 pp, 1978.

Albrecht, B. A.: Aerosols, cloud microphysics, and fractional cloudiness. Science, 245, 1227–1230, https://doi.org/10.1126/science.245.4923.1227, 1989.

Allen, M. R. and Ingram, W. J.: Constraints on future changes in climate and the hydrologic cycle. Nature, 419, 224–232, https://doi.org/10.1038/nature01092, 2002.

Andreae, M. O., Rosenfeld, D., Artaxo, P., Costa, A. A., Frank, G. P., Longo, K. M., and Silva-Dias, M. A.: Smoking rain clouds over the Amazon. Science, 303, 1337–1342, https://doi.org/10.1126/science.1092779, 2004.

Bretherton, C. S., Widmann, M., Dymnikov, V. P., Wallace, J. M., and Bladé, I.: The effective number of spatial degrees of freedom of a time-varying field. J. Clim., 12, 1990–2009, https://doi.org/10.1175/1520-0442(1999)012<1990:TENOSD>2.0.CO;2, 1999.

Cubasch, U., and Coauthors: Projections of Future Climate Change. Climate Change 2001: The Scientific Basis., J. T. Houghton and Y. H. Ding, Eds., Cambridge Univ. Press, Ch. 9, 524–582, 2001.

Dai, J., Xing, Y., Rosenfeld, D., and Xu, X. H.: The suppression of aerosols to the orographic precipitation in the Qinling Mountains. Chinese J. Atmos. Sci., 32, 1319–1332, 2008 (in Chinese).

Flato, G., and Coauthors: Evaluation of climate models. In Climate Change 2013: The Physical Science Basis. Contribution of Working Group I to the Fifth Assessment Report of the Intergovernmental Panel on Climate Change. Stoker, T. F., Qin, D., Plattner, G.-K., Tignor, M., Allen, S. K., Doschung, J., Nauels, A., Xia, Y., Bex, V. and Midgley, P. M. Eds. Cambridge University Press, pp. 741–882, https://doi.org/10.1017/CBO9781107415324.020, 2013.

Hansen, J., Sato, M., and Ruedy, R.: Radiative forcing and climate response. J. Geophys.

Res., 102, 6831–6864, https://doi.org/10.1029/96jd03436, 1997.

Koren, I., Martins, J. V., L. A. Remer, L. A., and Afargan, H.: Smoke invigoration versus inhibition of clouds over the Amazon. Science, 321, 946–949, https://doi.org/10.1126/science.1159185, 2008.

Liu, R., Liu, S. C., Cicerone, R. J., Shiu, C.-J., Li, J., Wang, J., and Zhang, Y.: Trends of extreme precipitation in eastern China and their possible causes, Adv. Atmos. Sci., 32(8), 1027–1037, https://doi.org/10.1007/s00376-015-5002-1, 2015.

Liu, R., Liu, S. C., Shiu, C.-J., Li, J., and Zhang, Y.: Trends of regional precipitation and their control mechanisms during 1979–2013, Adv. Atmos. Sci., 33(2), 164–174, https://doi.org/10.1007/s00376-015-5117-4, 2016.

National Research Council: Critical Issues in Weather Modification Research. The National Acedemies Press, Washington, D. C., USA, 143 pp, https://doi.org/10.17226/10829, 2003.

Ramanathan, V., Crutzen, P. J., Kiehl, J. T., and Rosenfeld, D.: Aerosols, climate, and the hydrological cycle. Science, 294, 2119–2124, https://doi.org/10.1126/science.1064034, 2001.

Sun, Y., Solomon, S., Dai, A., and Portmann, R. W.: How often will it rain?, J. Clim., 20(19), 4801–4818, https://doi.org/10.1175/JCLI4263.1, 2007.

Tao, W.-K., Chen, J.-P., Li, Z. Q., Wang, C. E., and Zhang, C. D.: Impact of aerosols on convective clouds and precipitation. Rev. Geophys., 50, https://doi.org/10.1029/2011rg000369, 2012.

Trenberth, K. E.: Atmospheric moisture residence times and cycling: Implications for rainfall rates and climate change, Clim. Change, 39(4), 667–694, https://doi.org/10.1023/A:1005319109110, 1998.

Trenberth, K. E., Dai, A., Rasmussen, R. M., and Parsons, D. B.: The changing character of precipitation, Bull. Am. Meteorol. Soc., 84(9), 1205–1217, https://doi.org/10.1175/BAMS-84-9-1205, 2003.

Warner, J. and Twomey, S.: The production of cloud nuclei by cane fires and the effect on cloud droplet concentration. J. Atmos. Sci., 24, 704–706, https://doi.org/10.1175/1520-0469(1967)024<0704:Tpocnb>2.0.Co;2, 1967.

Warner, J.: Smoke from sugar-cane fires and rainfall. In: Proceedings, International Conference on Weather Modification; Canberra. American Meteorological Society, 191–192, http://hdl.handle.net/102.100.100/317147?index=1, 1971.

---

## Author Response (AR3)

Dear Editor,

Thanks for your careful editing on our manuscript entitled "Observed Trends of Clouds and Precipitation (1983–2009): Implications for Their Cause(s)" (MS No.: acp-2020-577). We have accepted all those technical corrections and made some minor revisions. We really appreciate your kind help and comments of other two anonymous referees.